# Development of broadly neutralizing antibodies targeting the cytomegalovirus subdominant antigen gH

Andrea J. Parsons[1,2], Sabrina I. Ophir[1], J. Andrew Duty[1,3], Thomas A. Kraus[3], Kathryn R. Stein[1,2], Thomas M. Moran[1,3] & Domenico Tortorella [1✉]

Human cytomegalovirus (HCMV) is a β-herpesvirus that increases morbidity and mortality in immunocompromised individuals including transplant recipients and newborns. New anti-HCMV therapies are an urgent medical need for diverse patient populations. HCMV infection of a broad range of host tissues is dependent on the gH/gL/gO trimer and gH/gL/UL28/UL130/UL131A pentamer complexes on the viral envelope. We sought to develop safe and effective therapeutics against HCMV by generating broadly-neutralizing, human monoclonal antibodies (mAbs) from VelocImmune® mice immunized with gH/gL cDNA. Following high-throughput binding and neutralization screening assays, 11 neutralizing antibodies were identified with unique CDR3 regions and a high-affinity ($K_D$ 1.4-65 nM) to the pentamer complex. The antibodies bound to distinct regions within Domains 1 and 2 of gH and effectively neutralized diverse clinical strains in physiologically relevant cell types including epithelial cells, trophoblasts, and monocytes. Importantly, combined adminstration of mAbs with ganciclovir, an FDA approved antiviral, greatly limited virus dissemination. Our work identifies several anti-gH/gL mAbs and sheds light on gH neutralizing epitopes that can guide future vaccine strategies.

[1] Department of Microbiology, Icahn School of Medicine at Mount Sinai, New York, NY 10029, USA. [2] Graduate School of Biomedical Sciences, Icahn School of Medicine at Mount Sinai, New York, NY, USA. [3] Center of Therapeutic Antibody Development, Icahn School of Medicine at Mount Sinai, New York, NY 10029, USA. ✉email: domenico.tortorella@mssm.edu

Human cytomegalovirus is the largest β-herpesvirus with an ~235 kB dsDNA genome that encodes for >192 viral proteins and contains several miRNAs and ncRNAs[1–3]. HCMV establishes a life-long infection, maintaining a latent reservoir within the bone marrow[1] and viral shedding can occur after reactivation. The estimated global seroprevalence of HCMV is ~83%, with higher prevalence in lower socio-economic groups and women of childbearing age, the latter attributed to increased contact with young children[4,5]. Infections are typically asymptomatic in immuno-competent hosts but can cause severe, life-threatening complications in immunocompromised individuals or people with an immature immune system[6]. High-risk groups for HCMV include patients with AIDS, transplant recipients, and infants. Congenital HCMV infections occur upon reactivation or primary infection during pregnancy and subsequent transmission of the virus to the developing infant[7,8]. Congenital HCMV infection can be associated with severe birth defects including sensorineural hearing loss, microcephaly, and periventricular calcifications[7,9–12]. Despite numerous clinical trials to identify a safe and effective therapy to reduce transmission and/or disease severity in congenitally infected infants, none have yet to be approved, emphasizing the importance of developing unique strategies to treat HCMV[7].

Most of the current FDA-approved drugs for treatment of HCMV, including ganciclovir, valganciclovir, foscarnet, cidofovir, and letermovir, target late stages of the virus life cycle including viral replication and DNA packaging with varying efficacy and dose-related cytotoxicity[11,13,14]. Hyperimmune globulin (HIG) isolated from HCMV seropositive donors has been approved for the prophylaxis of HCMV disease associated with solid organ transplantation[15], but its use requires high doses, frequent IV administration, contains non-specific antibodies, and has lot-to-lot variability that impairs treatment efficacy[16–19]. Use of monoclonal antibodies to prevent infection and limit viral load has been evaluated previously and could provide several advantages over HIG[20–22].

The production of human antibodies in transgenic animals provides several advantages including in vivo affinity maturation, increased diversity, and clonal selection for antibody optimization[23]. Until recently, the large size of the human Ig loci presented a challenge in transgenic antibody technology. VelocImmune® mice were generated using an in situ, genetic humanization approach where 3 Mb segments of both the mouse heavy and light chains (κ) were replaced with their human counterparts[24,25]. The result is a transgenic mouse that generates human/mouse hybrid antibodies[25]. The human/mouse chimeric antibodies have a Fab-heavy and -light (κ) chain derived from human sequences and a mouse constant region to optimize interactions with B-cell receptors during B-cell development. Several monoclonal antibodies (mAbs) generated in VelocImmune® mice have been approved by the FDA; for example, Alirocumab targets PCSK9 for the treatment of hyperlipidemia, Dupilumab targets IL-4R to treat atopic dermatitis and asthma, Sarilumab is against IL-6R for treatment of rheumatoid arthritis and Cemiplimab targets PD-1 and has been approved for treatment of advanced cutaneous squamous cell carcinoma[23,26]. More recently, anti-SARS-CoV-2 antibodies raised against the Spike protein have been approved for emergency use to treat COVID-19[27–29]. These studies support the use of VelocImmune® mice to develop anti-HCMV neutralizing antibodies capable of limiting HCMV-associated diseases.

The enveloped virion surface is studded with HCMV protein and complexes, including gB, gM/gN, gH/gL/gO, and gH/gL/UL128/UL130/UL131A, that mediate entry into fibroblasts, epithelial, endothelial, and myeloid cells[1,30]. The gM/gN complex is the most abundant on the surface of HCMV virions and plays a key role in attachment through interactions with cell surface heparin sulfate proteoglycans; it also plays a role in viral dissemination. The envelope trimer complex (gH/gL/gO) and pentamer complex (gH/gL/ UL128/UL130/UL131A, PC) are mutually exclusive and dictate cell tropism. The trimer mediates entry into fibroblasts by interacting with the fusion protein gB at the cell surface in a pH-independent process[31], while the PC is essential for entry into epithelial, endothelial, and myeloid cells[32]. Targeting the gH protein shared by both complexes would effectively broadly limit infection. Component analysis of serum samples using adsorption studies shows the majority of neutralizing antibodies generated following a natural infection target the UL proteins of the gH/gL-complex suggesting that gH epitopes are subdominant[33–37].

Using sequential DNA immunization with a plasmid expressing gH/gL, a panel of 11 broadly neutralizing anti-gH-human monoclonal antibodies (mAbs) blocked infection in clinically relevant cell types and diverse virus strains. The specificity of the monoclonal antibodies to limit infection and their improved efficacy in combination experiments with ganciclovir support the potential of gH-specific antibodies as a therapeutic against HCMV.

## Results

### Identification of HCMV neutralizing antibodies from gH/gL immunized VelocImmune® transgenic mice.

To generate a panel of neutralizing, human monoclonal antibodies (mAbs) against HCMV, VelocImmune® mice were immunized with either a HCMV lab strain (Merlin), a combination of HCMV clinical strains (TB40/E and VHL/E), or plasmid encoding TB40/E gH/gL cDNA. The workflow for antibody discovery is outlined in Fig. 1a. Sera from immunized mice were assessed for neutralization capacity using a high-throughput neutralization assay (Fig. 1b) with the AD169 strain (BADrUL131-C4) that contains the UL131-UL128 ORF of HCMV TR and expresses GFP (denoted AD169R)[38]. Infection was examined at 18 hpi with CytoGam® and normal mouse serum (NMS) used as positive and negative controls, respectively. Incubation of virus with serum from gH/gL cDNA immunized mice resulted in a robust, dose-dependent neutralization and provided protection against the mismatched AD169R strain in both epithelial and fibroblast cells (Fig. 1b). Immunization with TB40/E and VHL/E produced antibodies capable of effectively neutralizing AD169R infection in epithelial cells with >90% neutralization at a 1:500 dilution in 3/5 mice (Fig. 1b). Surprisingly, mice immunized with Merlin displayed limited neutralization capacity compared to the other immunization strategies. The gH/gL cDNA immunization strategy was superior as an immunogen as demonstrated by the neutralization activity in both epithelial and fibroblast cell types.

In order to identify monoclonal antibodies (mAbs) capable of broadly inhibiting infection, we selected three mice (Mouse #29, 32, 33) with the highest neutralizing capacity from the gH/gL cDNA immunized group to create hybridoma clones. Collectively, 4298 hybridoma clones were screened using high-throughput flow cytometry to detect binding to 293ExpiF cells expressing gH/gL. We identified 161 clones with >2-fold mean fluorescence intensity (MFI) for gH/gL expressing cells. A heat map for each fusion summarizes the MFI for each clone (Fig. 1c). Binding positive clones were then screened to identify neutralizing antibodies against AD169R (Fig. 1d).

Based on binding to 293ExpiF-gH/gL cells and neutralization capacity, 24 clones sourced from all three fusions were sequenced and isotyped. Of these clones, 12 unique IgG antibodies (11

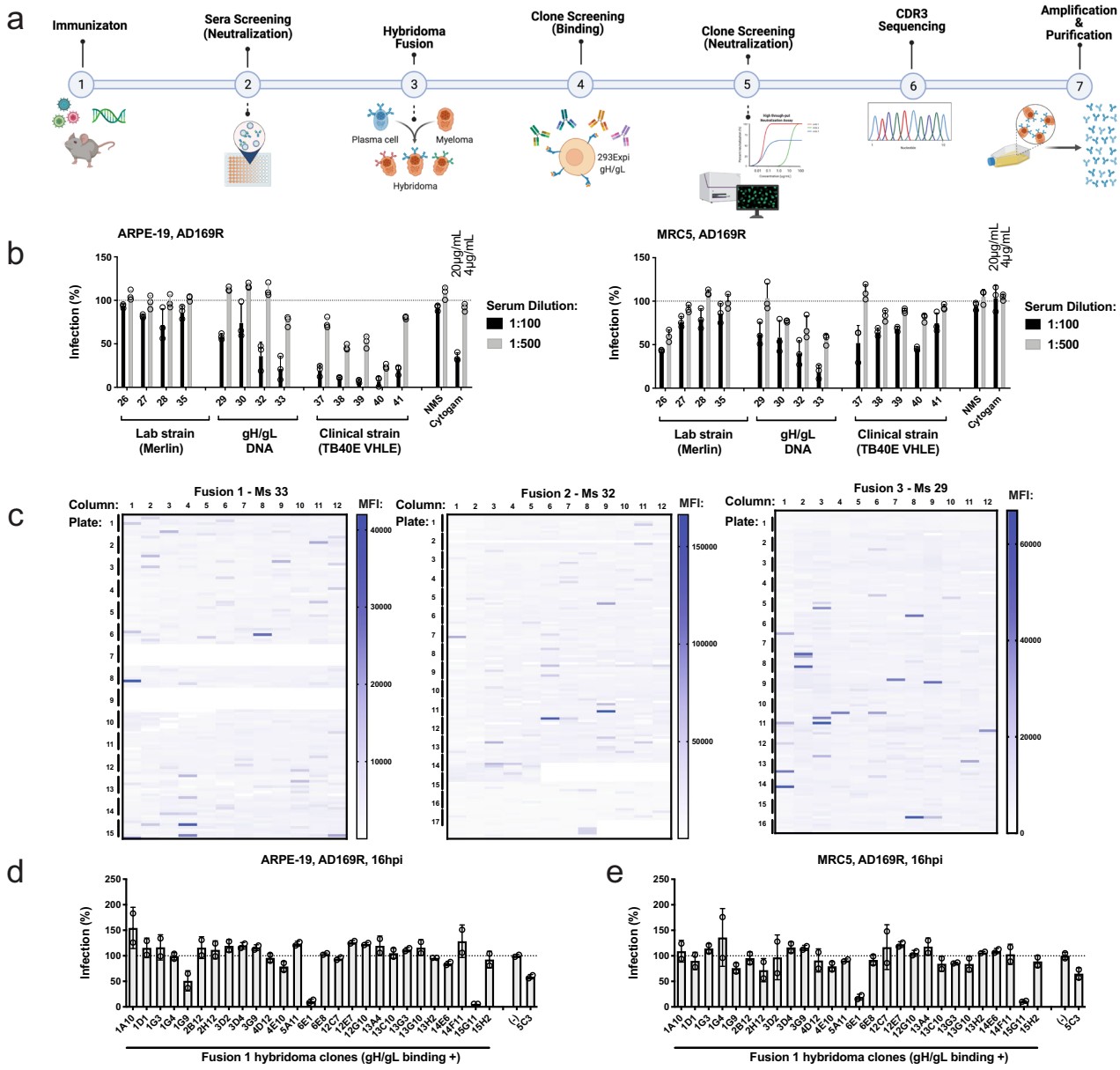

**Fig. 1 Sequential immunization elicits neutralizing antibodies identified through hybridoma screening. a** A schematic of immunization, hybridoma fusion and screening workflow to identify neutralizing monoclonal antibodies was created using Biorender Software. **b** Mice were repeatedly immunized with purified virus or plasmid encoding gH/gL, and blood was collected to evaluate the neutralization capacity of mouse sera in epithelial (ARPE-19) and fibroblast (MRC5) cells using reporter virus AD169R (MOI 0.1). Percent infection was quantified using GFP and normalized to virus incubated in the presence of normal mouse serum (NMS). Each serum condition was tested in technical triplicate ($n = 3$). **c** Antibody clones capable of binding gH/gL were identified following hybridoma supernatant incubation with 293ExpiF cells transiently expressing gH/gL. Mean fluorescence intensity (MFI) was determined for each sample and summarized by fusion ($n = 1$). Hybridoma supernatants from fusion 1 were screened for neutralization of AD169R in ARPE-19 (**d**) and MRC5 (**e**) cells and relative percent infection was normalized to virus infection only. The supernatant from each hybridoma was tested in technical duplicates ($n = 2$) during initial screening for both cell lines and dilutions. Error bars represent standard deviation from the mean.

neutralizers and 1 non-neutralizer) were expanded and purified for further characterization. Sequencing of the heavy chain from each hybridoma clone revealed diverse CDR3 lengths (10–20 aa) and found the *IGHJ6*02* gene to be predominant (50%) across the panel (Table 1). A phylogenetic tree was created based on the Jukes-Cantor model and displayed using the neighbor-adjoining method to determine the genetic diversity among the antibodies clustered into three major groups (Supplementary Fig. 1). The V-D-J allele and isotype for each antibody are summarized in Supplementary Fig. 1. In short, immunization with gH/gL cDNA elicited a broad,

robust neutralizing immune response, consisting primarily of IgG2a antibodies with diverse CDR3 regions.

**HCMV antibodies are broadly neutralizing and cross-protective**. The mAbs were next evaluated for their ability to limit infection in diverse cell types using a neutralization assay. AD169R (MOI 0.2) was preincubated with purified mAbs (0–50 μg/mL) prior to infection of epithelial cells, placental tissue-derived HTR-8/SVneo trophoblasts, and fibroblasts. All mAbs except 12H11 were able to demonstrate some level of protection in the

**Table 1 Antibody diversity, allelic frequency, and hypermutation of 12 monoclonal antibodies that bind gH.**

| Mouse ID | Antibody | Isotype | CDR3 length | AA junction | V gene and allele | V identity (%) | J gene and allele | J identity (%) | D gene and allele |
|---|---|---|---|---|---|---|---|---|---|
| Mouse 33 | 1G9 | IgG2a | 14 | CANHPNVLMIFVQDFW | IGHV5-51*01F | 96.88 | IGHJ6*02F | 85.48 | IGHD2-8*01F |
|  | 6E1 | IgG2a | 15 | CARRGYNFGYYYGMDVW | IGHV5-51*01F | 96.88 | IGHJ6*02F | 85.48 | IGHD5-18*01F |
|  | 15G11 | IgG2a | 10 | CARGGLGAFDIW | IGHV4-31*03For IGHV4-31*06F | 96.56 | IGHJ3*02F | 94 | IGHD7-27*01F |
| Mouse 32 | 1D11 | IgG2a | 15 | CAKSKHWGDYYYTMDVW | IGHV3-9*01F | 97.92 | IGHJ6*02F | 85.48 | IGHD4-17*01F |
|  | 9A12 | IgG2a | 11 | CARDPNWNFFDYW | IGHV1-18*01F | 97.92 | IGHJ4*02F | 97.92 | IGHD1-1*01F |
|  | 4E7 | IgG2a | 16 | CARSSGWYRNYSYGMDVW | IGHV3-48*03F | 95.83 | IGHJ6*02F | 90.32 | IGHD6-13*01F |
|  | 10F8 | IgG2a | 10 | CAREAYSNYGVW | IGHV3-33*01For IGHV3-33*06F | 95.83 | IGHJ6*03F | 78.18 | IGHD4-11*01 ORF |
|  | 12H11 | IgG2a | 17 | CARLGHVQFGYYYDMDVW | IGHV1-69*05F | 96.53 | IGHJ6*02F | 91.94 | IGHD3-16*01F |
| Mouse 29 | 10H6 | IgG2a | 20 | CARRGNWNPPYFYYRYNGLDVW | IGHV1-18*01F | 90.62 | IGHJ6*02F | 88.71 | IGHD1-1*01F |
|  | 11D3 | IgG2b | 13 | CARDSGSYSGFDYW | IGHV3-13*01F | 92.28 | IGHJ4*02F | 91.67 | IGHD3-22*01F |
|  | 13G1 | IgG2a | 15 | CARIDYSNYIGNWFDPW | IGHV2-26*01F | 97.25 | IGHJ5*02F | 96.08 | IGHD4-11*01 ORF |
|  | 14E1 | IgG2a | 10 | CARDHGFFFDYW | IGHV1-18*01F | 97.92 | IGHJ4*02F | 93.75 | IGHD1-26*01 |

The immunoglobulin isotype, heavy chain CDR3 length, and sequence for each antibody are summarized following Sanger sequencing. Allele frequency and percent identity to germline are provided for IgG V, D, and J genes after sequences were blasted against IMGT human databank of germline genes using V-QUEST analysis.

neutralization assays (Fig. 2a). The FDA-approved HIG CytoGam® was less efficacious in fibroblast infections due to enrichment of anti-UL128/UL130/UL131A antibodies[37]. Most of the mAbs were more effective at limiting infection of epithelial or trophoblast cells than fibroblasts indicating that the anti-gH mAbs have a greater effect on pentamer-based entry. Using PMA-differentiated THP-1 macrophages, mAbs 15G11, 9A12, and 13G1 consistently neutralized >90% infection of AD169R (Supplementary Fig. 2). The half-maximal inhibitory concentrations ($IC_{50}$, Table 2) for 15G11, 9A12, and 13G1 ($IC_{50}$: 0.01–7 µg/mL using 4-parameter non-linear regression analysis) demonstrate that these mAbs significantly limit infection of diverse cell types. In summary, we identified a panel of mAbs that block infection of diverse cell types and outperform hyperimmune globulin preparations currently used in the clinic.

We next evaluated the ability of the mAbs to neutralize HCMV strains TB40/E, TR, and Towne (Fig. 2b). These viruses represent HCMV strains isolated from North America and Europe between 1956 and 1999, with a gH/gL sequence identity >96.6%[39]. The mAbs 15G11, 9A12, and 13G1 again consistently neutralized infection of all strains, including AD169 in MRC5 cells (Fig. 2). Subtle variability in the gH/gL sequence across strains results in a loss of 4E7 efficacy against TR, while 10F8 was able to neutralize TB40/E and TR yet lost the ability to neutralize AD169R and Towne, suggesting differences in antibody-binding sites (Supplementary Fig. 2). Collectively, several mAbs broadly neutralized diverse HCMV strains in physiologically relevant cell types.

In addition to the standard $IC_{50}$ values, virus neutralization values at 50 and 2 µg/mL of antibody (Supplementary Fig. 3) further demonstrated the mAbs' effectiveness. The requirement for high concentrations of CytoGam® to neutralize infection (MOI 0.2) is blatantly clear as CytoGam® blocks only ~20–40% infection at 2 µg/mL on the other hand, select mAbs from our panel able to neutralize >60% of infection. Specificity of HCMV neutralization was confirmed by incubating antibodies with a reporter virus HSV-1 US-11eGFP as no significant changes in HSV-1 infection were seen compared to controls (Supplementary Fig. 4).

**The mAbs target HCMV gH/gL envelope protein complexes.** To confirm that mAbs bind to gH/gL complexes during infection, we assessed binding to infected ARPE-19 cells at 6 days post-infection (dpi) with AD169R (Fig. 3a). The percent infection was calculated based on GFP expression (x-axis) and the percent binding of each antibody was quantified using an anti-mouse antibody conjugated to AF647 (y-axis). While an increase in the GFP signal was observed by 6 dpi, the expression of gH/gL complexes at the cell surface was not absolute. However, binding can be detected in cells with the highest GFP MFI. Note, 12H11 had limited binding to infected cells and 4E7 bound to both mock and infected ARPE-19 cells. Higher binding (~40%) was detected when infected cells were fixed and stained to access binding to protein complexes in intracellular organelles (Supplementary Fig. 5a). These findings demonstrate that nearly all of the antibodies recognize gH/gL complexes in infected cells.

To confirm specificity for the HCMV gH/gL-dimer, trimer or pentamer, we used astrocytoma cells (U373) that constitutively expresses either gB, gH/gL, gH/gL/gO or gH/gL/UL128 (Fig. 3b). The gH/gL/UL128 is used as a surrogate for the pentamer complex[40]. The mAbs were incubated with either live or fixed/permeabilized cells to assess antibody binding as measured by MFI (Fig. 3b). The previously characterized anti-gH 5C3 and anti-gB 5A6 mAbs[40] were used as positive controls and an influenza-specific antibody (M2E10) was used as a negative control. Although most of the antibodies bound to gH/gL efficiently regardless of complex formation, the non-neutralizre

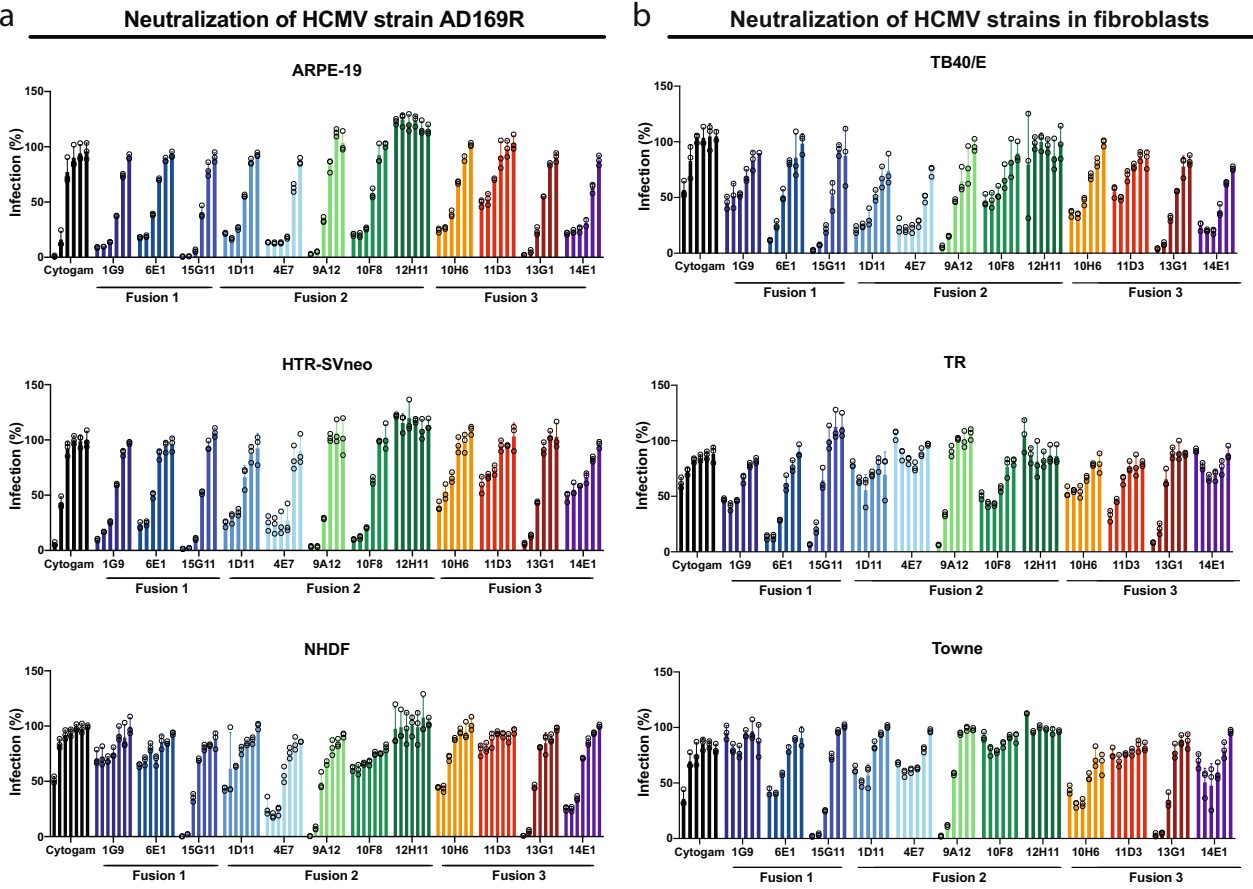

**Fig. 2 Monoclonal antibodies are broadly neutralizing. a** Increasing concentrations of monoclonal antibodies (μg/ml) were incubated with AD169R (MOI 0.2) to assess neutralizing capacity in epithelial cells, placental-derived trophoblasts and fibroblasts. Antibodies were diluted from 50 μg/mL using five-fold dilutions for epithelial and trophoblast cells. The mAbs were diluted using 10-fold dilutions from 100 μg/mL using the same AD169R (MOI 0.2) conditions in fibroblast neutralization assays. Percent infection was quantified using GFP expression at 18 hpi and normalized based on infection with virus only (no antibody). **b** The mAb panel was used to neutralize diverse HCMV strains (MOI 0.2) in MRC5 fibroblast cells using a five-fold serial dilution starting at 50 μg/mL. Relative percent infection at 18 hpi was determined following fixation and indirect immunofluorescence (IF) staining for IE1. All conditions were performed in technical triplicate ($n = 3$) and at minimum using $n = 2$ independent experiments. Error bars represent standard deviation from the mean.

mAb 12H11 and neutralizing mAb 1G9 showed reduced binding to intact gH/gL-dimer expressing cells and dramatically less binding to gH/gL/UL128-expressing cells. Interestingly, 4E7 is bound to permeabilized U373-gB cells suggesting it may bind to an intracellular protein; a result that is being pursued as an independent study. The 13G1 antibody bound non-specifically to the surface of U373-gB cells whose binding was inconsistent in fixed and permeabilized cells. A follow-up immunoprecipitation (IP) study using U373-gB cell lysate and antibodies 13G1 and 15G11 did not recover specific polypeptides. U373-gH expressing cells resulted in a loss of binding by 1G9, 10F8, 12H11, and 11D3 (Supplementary Fig. 5b) suggesting these mAbs recognize an epitope formed by gH interactions with gL. As expected, gH monomer is expressed intracellularly but does not result in high levels of gH at the cell surface, as gL is required for stability and trafficking[41]. We also confirmed the direct binding of the mAbs to gH/gL by IP using U373-gH/gL cell lysates (Fig. 3c) and confirmed weaker binding by 1G9 and 12H11 as previously seen by flow cytometry. The cellular localization of gH/gL complexes in TB40/E infected ARPE-19 cells (6 dpi) was confirmed for several antibodies including 15G11, which localized to the viral assembly complex (Fig. 3d). Collectively, this data confirm that the neutralizing antibodies recognize an epitope within the gH/gL-complex.

**Neutralizing antibodies bind two distinct regions of gH/gL**. In order to map the mAb's epitope, we initially performed a series of competition assays (Fig. 4a). A fixed amount of AlexaFluor 647 (AF647) labeled mAb (0.5 μg/mL) was mixed with increasing amounts of unlabeled mAb (0.1, 0.5, 1, or 5 μg/mL) prior to its addition to U373-gH/gL cells. The epitope of 5C3 was previously identified and is contained within the highly conserved region of gH (residues $_{481}$HTTERREIFI$_{490}$)[40]. The competition data suggest that 1D11, 4E7, 9A12, 10H6, 13G1, and 14E1 bind to a region in proximity to 5C3 (Fig. 4a, top panel). The loss of signal with unlabeled 5C3 acted as a positive control for competition. A competition assay using labeled 6E1 (Fig. 4a, middle panel) demonstrated that 10F8 and 11D3 compete with 6E1 for binding to the gH/gL complex, implying a distinct antibody-binding region from 5C3. To further investigate the binding specificity of 6E1, 10F8, and 11D3, we performed a competition study using labeled 10F8 (Fig. 4a, lower panel) and observed competition with 1G9, 15G11, and 11D3 but not 6E1. Interestingly, 14-4b a gH mAb[42,43] that recognizes a discontinuous epitope likely located near the membrane-proximal ectodomain of gH[44–46] competed with 10F8 implying they share a region of binding. Note, the non-neutralizing mAb 12H11 did not compete with the neutralizing mAbs suggesting that distinct regions within gH/gL are targets for neutralizing mAbs. The differences between 6E1 and 10F8 are

**Table 2 Half-maximal inhibitory concentration (IC$_{50}$) for Fusion 1–3.**

| Mouse ID | Fusion | Antibody | IC50 (ARPE-19) | | IC50 (HTR-svNeo) | | IC50 (MRC5/NHDF) | | AD169 | TR | Towne | IC50 (THP-1) | |
|---|---|---|---|---|---|---|---|---|---|---|---|---|---|
| | | | TB40E WT | AD169R | TB40E WT | AD169R | TB40E WT | AD169R | | | | TB40E WT | AD169R |
| - | - | Cytogam | 0.448 | 3.922 | 3.881 | 5.549 | >50 | >100 | >50 | >50 | 31.680 | 18.860 | 14.820 |
| Mouse 33 | 1 | 1G9 | 0.041 | 0.246 | 0.561 | 0.705 | 12.040 | >100 | >50 | 5.533 | >50 | 1.350 | 0.579 |
| | | 6E1 | 0.190 | 1.295 | 2.124 | 2.805 | 2.329 | >100 | 6.938 | 0.633 | 8.595 | 6.146 | 1.757 |
| | | 15G11 | 0.091 | 0.238 | 0.411 | 0.438 | 0.504 | 0.170 | 0.312 | 3.507 | 0.871 | <0.001 | 0.008 |
| Mouse 32 | 2 | 1D11 | 0.432 | 0.684 | 0.432 | 1.475 | 0.435 | 37.970 | 2.727 | >50 | 40.530 | 26.100 | 0.587 |
| | | 4E7 | 0.290 | 0.123 | 0.251 | 0.348 | 0.095 | 0.150 | 0.204 | >50 | >50 | 8.603 | 0.034 |
| | | 9A12 | 1.104 | 1.282 | 0.718 | 1.802 | 1.087 | 0.350 | 0.770 | 6.957 | 2.526 | 0.013 | 0.024 |
| | | 10F8 | 0.498 | 0.800 | 0.376 | 0.691 | 11.050 | >100 | 6.903 | 5.130 | >50 | 0.047 | 0.168 |
| | | 12H11 | >50 | >50 | >50 | >50 | >50 | >100 | >50 | >50 | >50 | 3.549 | >50 |
| Mouse 29 | 3 | 10H6 | 7.483 | 2.628 | 0.645 | 15.030 | 0.304 | 17.120 | >50 | 39.250 | 0.905 | >50 | 1.256 |
| | | 11D3 | 7.085 | >50 | 0.509 | >50 | 1.935 | >100 | >50 | 7.372 | >50 | >50 | 18.990 |
| | | 13G1 | 1.006 | 1.030 | 1.117 | 1.819 | 0.893 | 0.600 | 15.920 | 3.264 | 1.093 | 8.969 | 0.027 |
| | | 14E1 | 3.055 | 0.113 | >50 | 19.210 | 0.893 | 0.640 | >50 | >50 | >50 | ND | 0.035 |

The IC$_{50}$ for each neutralization assay is summarized in µg/mL for each antibody and organized by cell type and HCMV strain.

likely due to differences in affinity (Table 3). The K$_D$ was determined for each mAb (1.41–64.60 nM) and 10F8 binds to the HCMV pentamer with a higher affinity than 6E1. Collectively, these data suggest that the neutralizing mAbs interact with high affinity to two distinct regions of gH.

Further, 10F8, 1G9, and 11D3 do not bind to the gH monomer suggesting they may bind near the gH/gL interface or bind to a region of gH that is altered by subsequent binding of gL. Interestingly, none of the gH antibodies disrupted the binding of MSL-109, a neutralizing gH-human antibody shown to stabilize the gH/gL/gO-PDGFRα complex[47]. This high-resolution cryo-EM map revealed critical residues required for the interaction between MSL-109 and gH and confirmed the conservation of the gH/gL structure between trimer and pentamer complexes. Together, conserved regions of gH are effective targets of antibodies to block virus infection.

**Characterization of the neutralizing mAbs epitope.** To further define the gH epitopes of the anti-gH antibodies, we used a cyclic peptide microarray platform (PEPperCHIP) that defined the epitope of 5C3 and 10C10 [40]. This assay utilizes a peptide library consisting of overlapping peptides of 7, 10, and 13 aa of gH to identify an antibody epitope. Clones 1D11, 9A12, 10F8, and 15G11 were analyzed because they represent the two binding profiles to gH (Fig. 4a). The microarray binding profile of 9A12 to specific peptides suggests the antibody's epitope (Fig. 4b). The binding profile of 1D11, 9A12, 10F8, and 15G11 identified both unique and common peptides that interact with the different mAbs (Supplementary Fig. 6). In order to gain an appreciation of the potential binding region of the mAbs, previously published anti-gH epitopes were mapped onto the PDB 5VOB structure of gH/gL (Fig. 4c) alongside the predicted epitopes of the mAbs (Fig. 4d). The 1D11 antibody bound two peptides within Domain-2 of gH, similarly to 5C3 but localized to the opposite side of the protein (Fig. 4c, blue). Interestingly, 9A12, 10F8, and 15G11 consistently interacted with the peptide $_{187}$HRPHF$_{191}$ found in Domain-1. In addition, 9A12 and 15G11 bound strongly to $_{426}$LSKQNQQHLIPQW$_{438}$ located in Domain-2. The PEPperCHIP results predicted binding to 2–3 distinct regions of gH. One region lies within the alpha helix-rich domain, near a cleft that is predicted to interact with gB and on the opposite face of the gH molecule proposed to interact with a host receptor. Antibody binding to either region would result in HCMV neutralization by disrupting key gH functions.

To further characterize mAb binding to gH, a binding assay was performed with the anti-gH mAbs to gH alanine mutants predicted to disrupt binding. The binding studies were performed in BHK cells co-transfected with wildtype or mutant gH-AA and gL cDNA. Co-transfection with gL was required to increase gH protein stability and trafficking to the cell surface[41]. The anti-gH mAbs conjugated to AlexaFluor647 were evaluated for binding to gH(AA)-transfected BHK cells using a plate-based imaging cytometer. The relative binding (%) of the antibodies to wildtype gH and gH-AA mutants was calculated to characterize mAb binding profiles (Supplementary Fig. 6). Note, clone 11D3 could not be evaluated due to low labeling efficiency. The anti-gH antibodies MSL-109 and 5C3 were used as positive controls, while the negative controls were anti-BKV and anti-MHC class I (W6/32) antibodies. Antibody binding was examined against 12 alanine mutants covering five regions (aa.188–192, aa.316–318, aa.332–338, aa.433–438, and aa.534–537) spanning gH Domains 1 and 2. The gH-318AA mutant abrogated binding across the entire panel suggesting these residues are important for gH folding. Even though the exact epitopes of the anti-gH mAbs were not defined, the mAb binding profiles to the gH-AA constructs defined two major binding regions with additional specificity of some antibodies.

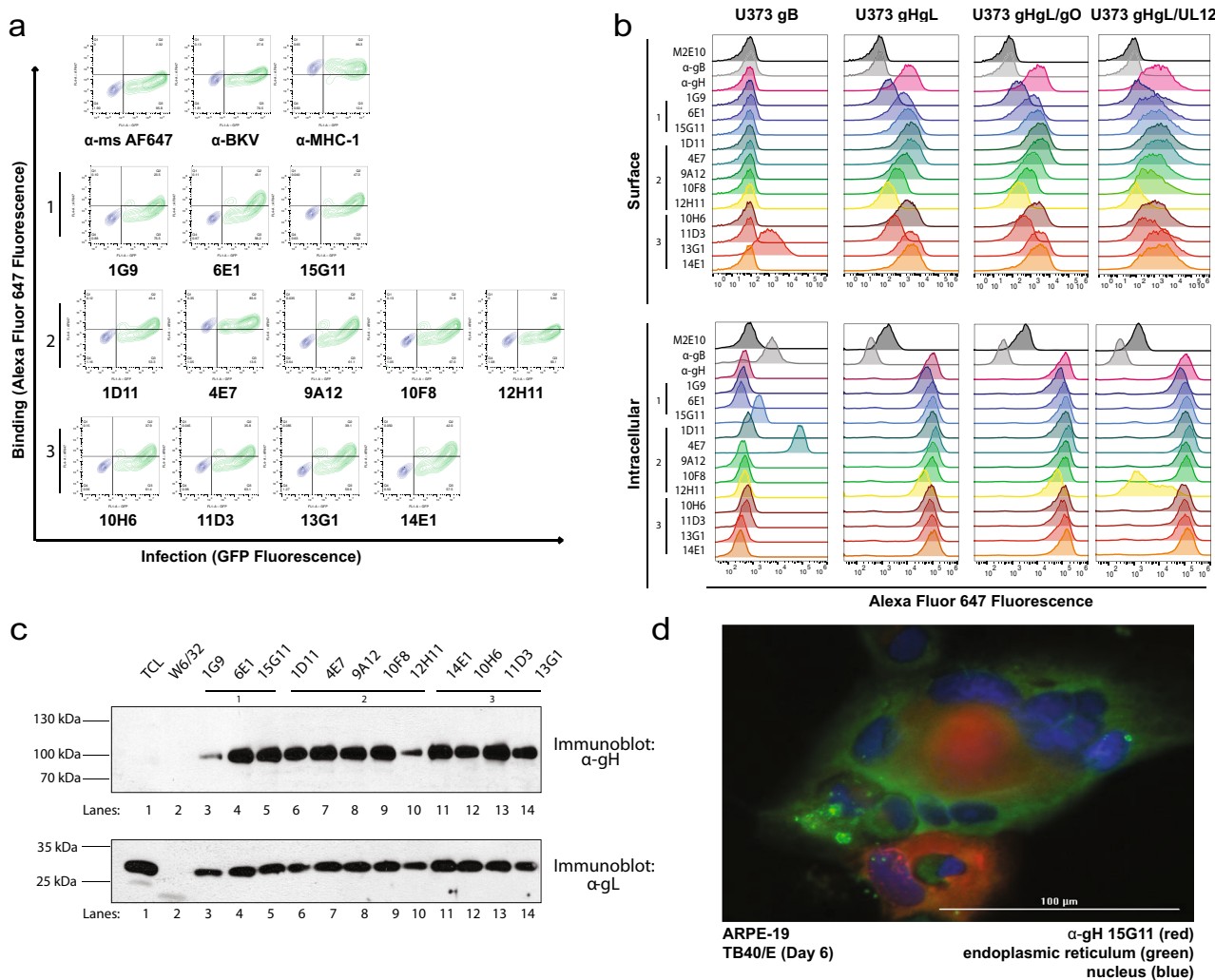

**Fig. 3 Monoclonal antibodies bind conformational epitopes on gH. a** ARPE-19 cells were infected with AD169R reporter virus (MOI 0.1) that expresses GFP (x-axis). At 6 dpi, cells were trypsinized, fixed, and permeabilized for staining with antibodies from Fusion 1–3 (2 µg/mL) and binding was quantified by flow cytometry using anti-mouse AF647 detection antibody (y-axis), $n = 3$ technical triplicates for an $n = 2$ experimental duplicates. **b** U373 astrocytoma cells expressing HCMV glycoproteins gB, gH/gL, gH/gL/gO, or gH/gL/UL128 were assessed for antibody (2 µg/mL) binding to cell surface or intracellularly from fixed and permeabilized cells using $n = 3$ technical triplicates and repeated, $n = 2$ experimental replicates. Note, M2E10 shows some cross-reactivity with the fixed and permeabilized cells. **c** The gH/gL-complex was isolated by immunoprecipitation (IP) using 5 µg of indicated mAb from U373-gH/gL cell lysates. U373-gH/gL total cell lysate (TCL) and an IP using W6/32 (α-MHC-I) antibody were used as controls. The TCL and IP ($n = 1$ lane per antibody) were resolved on a reducing SDS-polyacrylamide gel and subjected to an anti-gH and anti-gL immunoblot. **d** ARPE-19 cells infected with TB40/E WT were stained with 15G11 (α-gH, AlexaFluor 647), G1/296 (α-CLIMP63, AlexaFluor 488), and Hoechst reagent (0.01 µg/mL) 6 dpi to characterize gH localization within infected cells.

Interestingly, gH-334AA disrupted binding for all antibodies except 9A12 and 13G1 supporting the unique binding profiles among the most potent neutralizers in the panel. The anti-gH neutralizing mAbs likely bind to a discontinuous, conformational epitopes within Domains 1 and 2 of gH.

**Inhibiting HCMV infection with α-gH antibodies limits viral dissemination.** Next, we utilized a fluorescence-based virus plaque assay to evaluate the ability of the mAbs to block infection and prevent virus dissemination (Fig. 5). Using a Celigo Imaging Cytometer, multiple rounds of virus proliferation in live cells can be evaluated by collecting brightfield and GFP images along an infection time course with the GFP reporter virus AD169R. On Day 10 or 14, the cells were fixed and subjected to immunostaining for quantification of total cells or infection using GFP expression as a readout for infection. These images show the

formation and spread of HCMV multinuclear bodies over time and can then be quantified using 10,000 µm² as a cutoff for a viral plaque. Briefly, ARPE-19 and NHDF (Fig. 5a, b) cells were infected with AD169R (MOI 0.01 and MOI 0.001) in the presence of neutralizing antibodies (10 or 0.5 µg/mL) before both virus and antibody were removed and cells were overlaid with 1% SeaPlaque Agarose. Images were taken on Day 7 post-infection and the relative number of plaques/well was determined and compared to wells that received the irrelevant control antibody M2E10. The anti-gH mAb 5C3 and Cytogam® were used as positive controls. The neutralizing anti-gH mAbs outperformed CytoGam® in limiting plaque formation in both epithelial and fibroblast cells but were more effective at limiting HCMV infection in epithelial cells. These results are likely due to the diverse roles of gH/gL-complexes during infection of the two cell types. Consistent with previous data, 15G11, 9A12, and 13G1 were the most effective neutralizers. Of note in AD169R-infected fibroblasts, 1D11, 4E7,

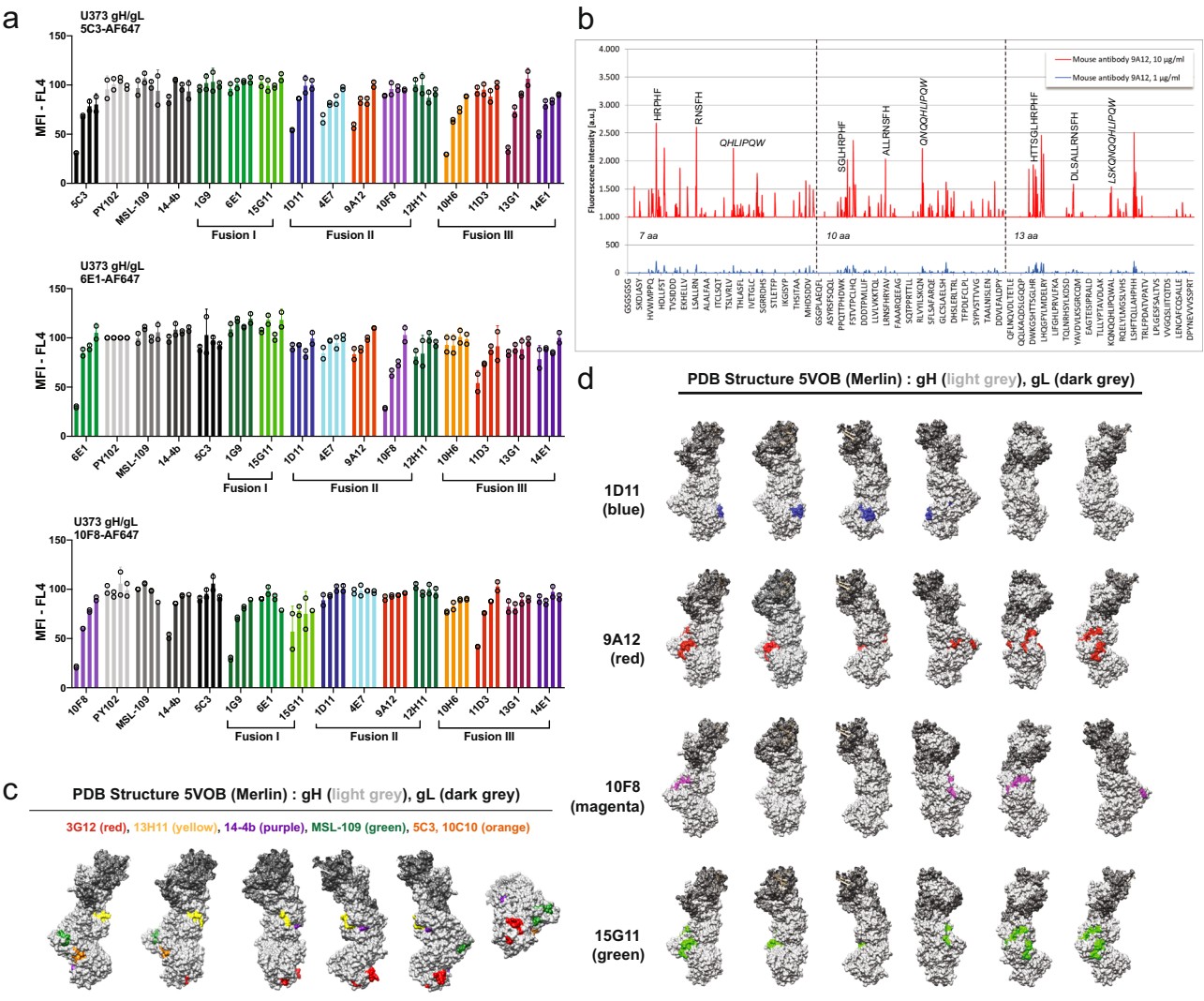

**Fig. 4 The anti-gH antibodies bind two distinct regions of gH. a** Competition assays were performed with U373-gH/gL cells incubated with increasing concentrations of unlabeled antibody (5–0.1 μg/mL, x-axis) and a constant amount (0.5 μg/mL) of AF647 labeled 5C3 (top), 6E1 (middle), and 10F8 (bottom). The relative MFI of AF647-positive cells is depicted after normalization to the average MFI of labeled antibody incubated with the irrelevant influenza antibody PY102. The full set of antibodies was tested with $n = 2$ technical triplicates and standard deviation from the mean is shown. **b** Representative plot of antibody 9A12 binding to the overlapping peptide libraries of 7, 10, and 13 aa in length corresponding to the gH aa sequence. **c** The model structure of gH/gL (PDB: 5VOB) is depicting previously reported antibody epitopes as indicated. **d** Mapping of proposed epitopes based on PEPperPRINT microarray results for each antibody on gH/gL structure.

and 12H11 reduced viral dissemination. This data represents the number of plaques >10,000 um², but the same trends were observed when examining the total number of infected cells per well. These data further support that anti-gH mAbs can effectively limit HCMV dissemination.

**α-gH antibodies do not block virus attachment**. To measure HCMV virion attachment and entry, we used a recombinant TB40/E that expresses GFP fused to the C-terminus of the tegument protein pUL32 (TB40/E UL32-eGFP) (Supplementary Fig. 7). The tegument protein is tightly associated with the capsid and allows for visualization of virus entry by examining GFP fluorescence intensity[48]. TB40/E UL32-eGFP was incubated with 15G11, 6E1, or the irrelevant antibody α-BKV at 10 μg/mL for 1 h at 4 °C. The virus/mAb mix was added to ARPE-19 cells followed by a citrate buffer (pH 3.2) wash at 0, 30, 60, or 120 min and then images were acquired to visualize the GFP virions (Supplementary Fig. 7a). We observed an increase in GFP virions

over the 120 min time course representing cell-bound virions. Representative images of the virions demonstrate that the neutralizing antibodies 6E1 and 15G11 do not dramatically alter the number of virions bound to the cells when compared to the α-BKV antibody, indicating that antibody neutralization likely occurs at a post-attachment step.

Further, we performed time-of-addition assays using the reporter virus AD169R. The neutralizing mAbs were added to AD169R-infected ARPE-19 cells at 0, 30, 60, and 120 min post-infection and then analyzed for infection by GFP-positive cells at 18 hpi (Supplementary Fig. 7b). Most of the neutralizing mAbs were able to inhibit infection for up 60 min post-infection supporting the model that gH mAbs target HCMV entry at a post-attachment step.

**Effectiveness of anti-gH antibodies in combination with ganciclovir.** Combination therapy targeting different steps of a virus life cycle have demonstrated to effectively limit virus dissemination,

**Table 3 Antibody-binding affinity to HCMV recombinant pentamer.**

| mAb | KD (nM) | kon(1/Ms) | kdis(1/s) | Full X² | Full R² |
|---|---|---|---|---|---|
| 5C3 | 7.64 | 1.42E + 04 | 1.09E-04 | 0.4975 | 0.9993 |
| 1G9 | 19.2 | 2.29E + 04 | 4.40E-04 | 0.5376 | 0.9977 |
| 6E1 | 16.9 | 5.40E + 04 | 9.11E-04 | 1.9191 | 0.9944 |
| 15G11 | 2.28 | 3.82E + 04 | 8.69E-05 | 1.6738 | 0.9986 |
| 1D11 | 5.66 | 8.23E + 03 | 4.66E-05 | 0.0561 | 0.9999 |
| 4E7 | 8.08 | 2.13E + 04 | 1.72E-04 | 0.2226 | 0.9997 |
| 9A12 | 3.47 | 2.48E + 04 | 8.60E-05 | 0.2753 | 0.9992 |
| 10F8 | 4.91 | 2.59E + 04 | 1.27E-04 | 0.2763 | 0.9997 |
| 12H11 | 15.9 | 2.44E + 04 | 3.88E-04 | 0.1464 | 0.9854 |
| 10H6 | 13 | 7.57E + 03 | 9.82E-05 | 0.1841 | 0.9993 |
| 11D3 | 1.41 | 1.17E + 04 | 1.65E-05 | 0.3144 | 0.9996 |
| 13G1 | 5.05 | 1.29E + 04 | 6.51E-05 | 0.113 | 0.9998 |
| 14E1 | 64.6 | 2.44E + 04 | 1.58E-03 | 0.9419 | 0.9923 |

Each $K_D$ was calculated based on antibody binding to gH/gL/UL128/UL130/UL131A pentamer (strain VR1814) using biolayer interferometry assays. $K_D$ values (nM) were calculated as a ratio of $k_{dis}/k_{on}$ and the resulting $R^2$ values are given for each antibody to determine fit.

prevent the rise of drug-resistant strains, and limit toxic effects associated with high concentrations of monotherapy[11,49–51]. Ganciclovir is a nucleoside analogue of 2'-deoxyguanosine used to prevent and treat HCMV infections in transplant recipients and individuals with AIDS[52] leading to reduced virus production[53]. Unfortunately, ganciclovir can cause hematological toxicity and neurotoxicity, and drug-resistant strains upon long-term treatment[14]. Thus, combination therapies would be an effective and safe therapeutic strategy against HCMV[51].

To test the effectiveness of anti-gH antibodies in combination with ganciclovir, we performed a fluorescence-based virus plaque assay in AD169R-infected ARPE-19 cells with a ganciclovir concentration below the $IC_{50}$ (2.5 μM) and increasing concentrations of anti-gH antibody (Fig. 6). M2E10 and CytoGam® were used as negative and positive controls respectively. A dramatic decrease in plaque number was observed in ganciclovir/antibody-treated cells on day 12 post-infection (Fig. 6 and Supplementary Fig. 8) and representative images for Fusion 1 and 2 mAbs (Fig. 6) and Fusion 3 mAbs (Supplementary Fig. 8) highlight the difference in plaque numbers and plaque size. The relative number of plaques on Day 7 (Supplemental Figure 8) and day 12 post-infection (Fig. 6b) were normalized based on the infection without antibody (black bars) or in the presence of the irrelevant antibody M2E10. Cells treated with ganciclovir (gray bars) demonstrated only a slight reduction of plaques at 12 dpi. In all cases, the combination treatment decreased plaque number at all mAb concentrations. Again, the anti-gH mAbs outperformed CytoGam® alone and in combination with ganciclovir. Strikingly, 15G11, 9A12, and 13G1 when combined with ganciclovir resulted in a >90–95% reduction in plaques at 10 μg/mL and 75% reduction at 2.5 μg/mL. Note, the day 12 plaques from M2E10 treatment created large plaques that merged making quantification difficult and resulting in a likely underestimate of relative plaque number. The non-neutralizing antibody 12H11 showed no significant reduction in infection. These findings suggest that the neutralizing antibody/ganciclovir combination would be an effective HCMV therapeutic strategy.

We then evaluated the synergy of select anti-gH mAbs with other anti-gH- or anti-gB-mAbs (Supplementary Fig. 9a, b). The anti-gH combinations did not improve inhibition suggesting they may block the same step in HCMV entry. When gH mAbs are combined with the neutralizing anti-gB antibody ITC-88[54], we observed no significant differences in neutralization capacity but at high concentrations of antibody the virus infection was completely blocked. Four mAbs (15G11, 9A12, 13G1, and 14E1)

were selected for future in vivo studies, the mouse Fc region of these mAbs was replaced with the human IgG1 Fc region. To ensure that the Fc domain did not impact their neutralization capacity, we performed a neutralization study (Supplementary Fig. 9c and Supplementary Table 1). The completely human 13G1 and 14E1 improved $IC_{50}$, 9A12 was unaffected and the $IC_{50}$ of 15G11 increased from 0.17 μg/mL to 0.37 μg/mL. These findings show that antibodies targeting gB and gH/gL complexes have a similar impact on virus entry and provide additional evidence that fully human antibodies developed from transgenic mice maintain neutralization capacity after conversion of the Fc region. This information is important, as in vivo studies for the top neutralizers are being pursued using a SCID mouse model.

## Discussion

Antibodies with fully human CDR3 regions isolated from transgenic animals represent a rapidly growing sector for antibody-based therapeutics[23,55]. The first antibody-based therapeutic derived from transgenic mice was approved by the FDA in 2006 for the treatment of metastatic colorectal cancer[56]. In the 16 years since 19 mAbs have been approved including four generated in VelocImmune® mice. Monoclonal antibodies with human Fab domains from transgenic mice have several advantages including the selection of high-affinity immunoglobulins, reduced production time, high-throughput methods for joining the variable regions to diverse human Fc domains, and implementation of diverse immunization strategies[23,55]. In this study, we generated a panel of broadly neutralizing anti-HCMV antibodies in VelocImmune® mice that target the HCMV envelope protein gH. To the best of our knowledge, this panel represents the first time VelocImmune® technology has been utilized to generate HCMV-specific antibodies and demonstrates the value of a directed immunization approach for specific HCMV immunogens.

The broad neutralization of anti-gH antibodies implies that effective anti-HCMV biologics can be developed to limit viral infection. HCMV HIG is approved as a prophylactic treatment for solid organ transplant recipients and is being evaluated for congenital infection. Clinical trials testing the safety and efficacy of HIG in congenital transmission have yet to yield consistent results[57]. A reduction of congenital HCMV following HIG treatment was observed in several trials[58–61]. However, a placebo-controlled, randomized trial in women treated monthly with CytoGam® beginning at the second trimester did not significantly prevent congenital infection[62]; a result consistent with a previous clinical trial[58,63]. The inconsistency of clinical trials may be due to varying treatment doses, frequency of administration, and trimester of HCMV infection indicates the need for comprehensive studies to optimize treatment conditions to prevent congenital HCMV.

Replacing HIG with specific, highly neutralizing mAb cocktails could reduce dose and frequency of administration, eliminate product variability, reduce off-target effects, and increase efficacy. Monoclonals against gH complexes and anti-gB mAbs have been evaluated in Phase I safety trials[22,64] and a Phase 2 trial of RG7667 (anti-gH and anti-PC antibody combination) in high-risk kidney transplants recipients showed delayed-kinetic to HCMV viremia and less HCMV disease[21]. Yet, a Phase II clinical trial with CSJ148 (anti-gB and anti-PC antibody combination) did not meet the primary efficacy endpoint in preventing HCMV reactivation in HSCT recipients[65]. However, CSJ148-treated patients had decreased viral loads, increased time for pre-emptive therapy, and a shorter median duration of pre-emptive therapy. These studies imply that anti-HCMV mAbs can effectively limit viral dissemination, but more testing of mAbs cocktails and mAb/inhibitor combinations need to be optimized to prevent virus dissemination and reactivation. Recently published serological profiling of six subjects

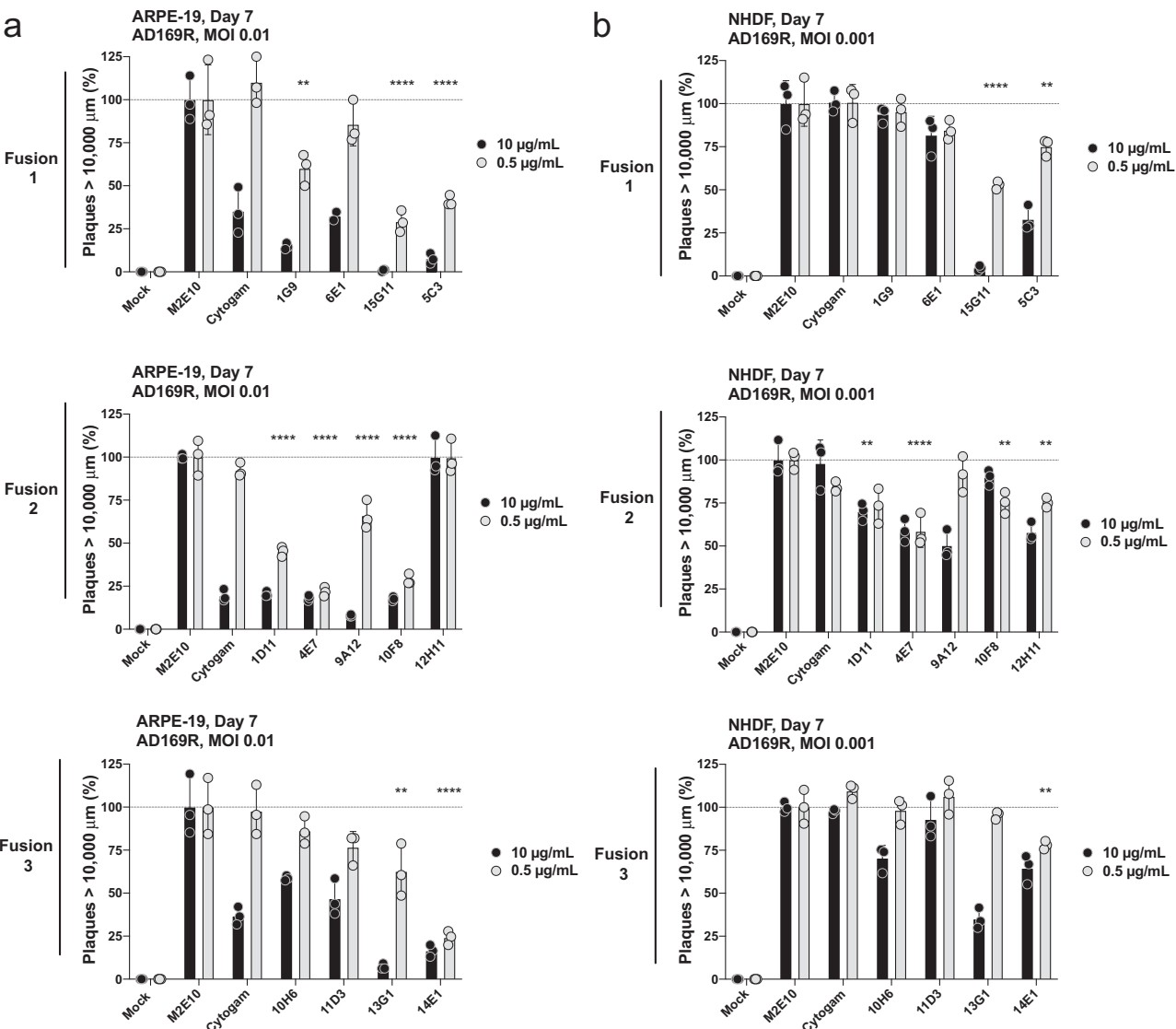

**Fig. 5 Anti-gH antibodies significantly reduce plaque formation in infected cells.** The relative number of focus forming units (plaques) (>10,000 μm²) on day 7 post-infection in both ARPE-19 (**a**) and NHDF (**b**) are shown for each antibody at 10 and 0.5 μg/mL using M2E10 (α-IAV) and Cytogam® as negative and positive controls per plate. The % of plaques was determined using single colony verification with 10,000 μm² as the minimum colony area based on GFP expression in infected cells. All conditions were repeated using $n = 3$ technical replicates and statistical significance was determined using Ordinary one-way ANOVA with multiple comparisons using Dunnett's correction with single pooled variance comparing the relative number of plaques at 0.5 μg/mL of gH mAb to plaques in M2E10 treated wells. Statistical significance is denoted as *$p < 0.05$; **$p < 0.01$; ***$p < 0.001$; ****$p < 0.0001$. Error bars represent standard deviation from the mean.

post-vaccination with the V160 replication-defective HCMV vaccine candidate identified a panel of 272 anti-gH neutralizing antibodies[66–68]. These studies support the role of neutralizing gH antibodies in providing protection against HCMV and the utilization of anti-gH mAbs as a therapeutic.

Serological profiling following natural HCMV infection shows that non-neutralizing and neutralizing antibodies are directed to viral envelope proteins[37,69–71]. Naturally occurring anti-gH antibodies (IgG1) have yet to demonstrate ADCC activity, yet anti-gB-antibodies can induce ADCC and ADCP, activities associated with protection. In a Phase II clinical trial, vaccination with gB/MF59 reduced HCMV transmission in young women by 50% but antibody titers peaked at 6.5 months post-vaccination and then rapidly waned[72]. The development of effective HCMV therapeutics should include a focused approach on antibody engineering and optimizing mAbs combinations in addition to

unique mAb discovery. Modifications of the mAb sequence can improve the half-life of passively administered mAbs or to improve FcγR and FcRn binding has been applied to several antibodies to elicit or increase existing effector functions including ADCC, ADCP, and C1q binding to initiate complement-dependent cytotoxicity[73,74].

In this study, we demonstrate that the combination of anti-gH neutralizing mAbs with ganciclovir significantly limited virus dissemination (Fig. 6) supporting the use of mAbs in combination with small molecules inhibitors. In a recently published case study, a combination of letermovir and HIG was used to successfully treat a ganciclovir-resistant strain of HCMV in a renal transplant recipient[75] giving credence to the therapeutic paradigm of mAb/drug combinations. Combination therapy may lower the required dose of antiviral drugs such as ganciclovir and letermovir and limit the associated toxicity and emergence of viral

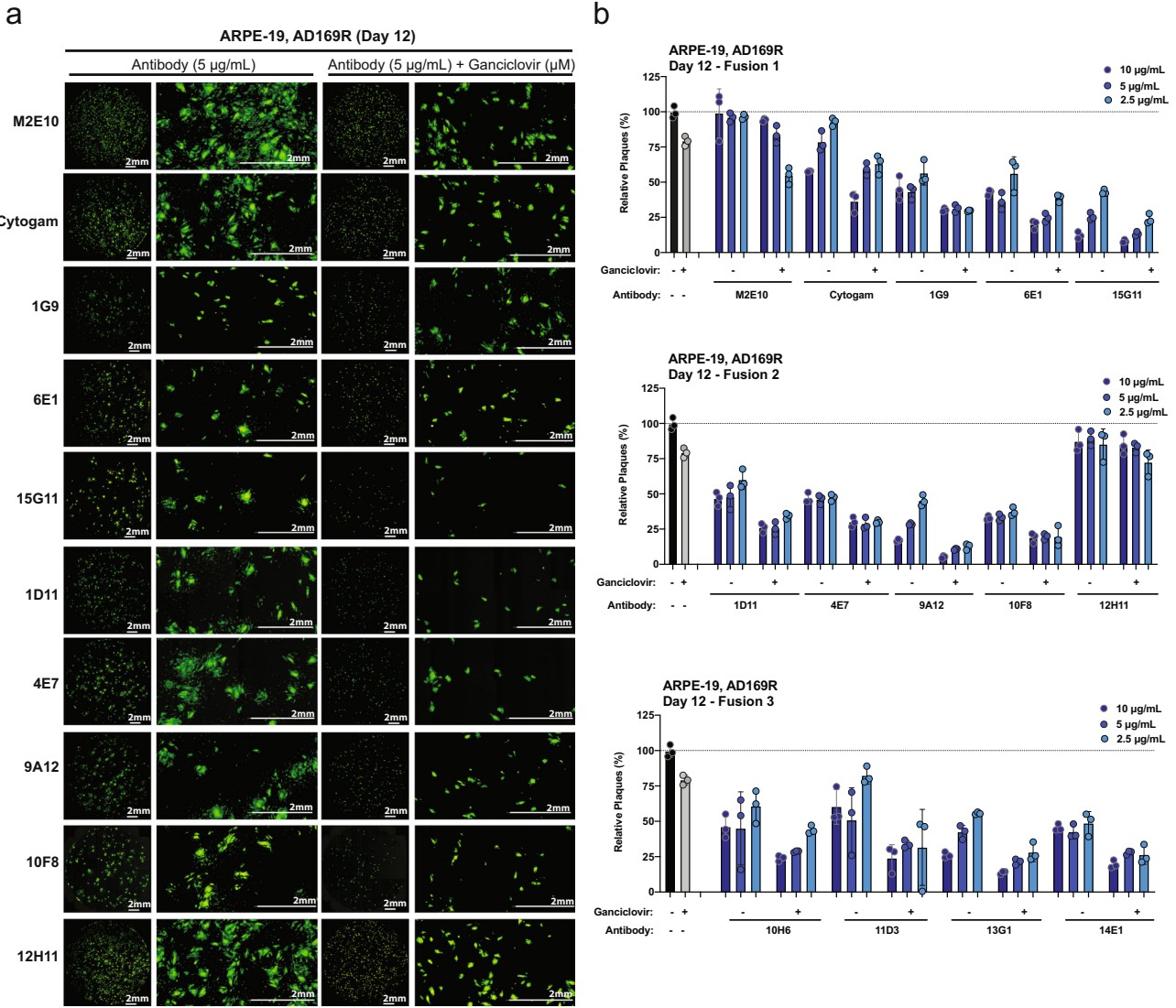

**Fig. 6 Anti-gH antibodies with ganciclovir reduces viral plaques. a** Representative whole well and zoomed GFP images of AD169R-infected ARPE-19 cells treated with M2E10, Cytogam®, Fusion 1 and Fusion 2 antibodies ± ganciclovir infected at 12 dpi. **b** Relative number of plaques at 12 dpi in ARPE-19 cells infected with AD169R cells treated with M2E10, Cytogam®, and Fusion 1, 2, and 3 antibodies ± ganciclovir. Infection was performed in the presence of antibody at three different concentrations ± ganciclovir (2.5 μM) with *n* = 3 technical replicates per condition. At 12 dpi, from the GFP images, the plaque numbers were calculated relative to M2E10. Each condition was completed in triplicate and 10,000 μm$^2$ was used a cutoff for plaque area. Error bars represent standard deviation from the mean.

mutations, as a combination approach would be expected to effectively limit virus replication and associated diseases. In conclusion, the data support further development of the broad-neutralizing mAbs as potential therapeutics to limit HCMV infection and dissemination.

## Methods

**Cell lines, antibodies, and viruses**. The MRC5 lung fibroblasts (ATCC, CCL-171), normal human neonatal dermal fibroblasts (NHDF, Lonza, CC-2509), BHK (ATCC, CCL-10), and the U373 astrocytoma cell lines (a gift from Dr. Hidde Ploegh, Boston Children Hospital, Harvard Medical School) were cultured in Dulbecco's modified Eagle's medium (DMEM, Corning, 10-013-CV). The ARPE-19 human retinal epithelial cells (ATCC, CRL-2302) were cultured in DMEM and F-12 medium (Gibco, 11765-054) mixed at 1:1 ratio. The trophoblast cell line, HTR-8/SVneo (ATCC, CRL-3271) were obtained and cultured in RPMI medium (Corning, 10-041-CV). The DMEM, RPMI, and DMEM/F-12 mediums were all supplemented with 10% fetal bovine serum (FBS), 1 mM HEPES (Corning, 25-060-CI), 100 U/mL of penicillin, and 100 g/mL of streptomycin (100X Pen/Strep, Corning, 30-002-CI). The U373 cells that constitutively express the HCMV

glycoproteins gB, gH/gL, gH/gLg/O, and gH/gL/UL128 were generated as previously described[40]. All cell lines were kept at 37 °C with 5% CO$_2$.

The antibodies M2E10 (anti-IAV) and PY102 (anti-IAV) were provided as gifts from Dr. Thomas Moran, Icahn School of Medicine and CytoGam® was purchased from CSL Behring LLC, NDC-44206-532-90. Monoclonal antibody W6/32 (anti-MHC-I) was a gift from Dr. Hidde Ploegh (Harvard Medical School) and the monoclonal antibody 14-4b (anti-gH) was a kind gift from William J Britt (UAB). The variable genes of the heavy and kappa chains of MSL-109 were synthesized based on published sequence (US Patent #5750106) as Gene Blocs (IDT, Coralville, IA) and cloned into the pFuse vector containing human constant IgG1 and kappa chains, respectively (InvivoGen, San Diego, CA). MSL-109 heavy and light chain IgG were co-transfected in HEK-293 cells and the antibody was purified from the cell supernatant. The polyclonal anti-gL immunoglobulins were generated in rabbits against the HCMV TB40/E gL peptide (aa. 265–278, PAHSRYGPQAVDAR). The following antibodies were purchased commercially; mouse anti-glyceraldehyde-3-phosphate dehydrogenase (GAPDH, EMD Millipore, MAB374), donkey anti-rabbit IgG conjugated to horseradish peroxidase (HRP) (Invitrogen, A16035) and donkey anti-mouse-HRP (Invitrogen, A16017). The secondary antibodies used for detection in flow cytometry experiments were goat-anti-mouse AlexaFluor 647 (AF647) (Invitrogen, A21236), chicken anti-rabbit AF647 (Invitrogen, A21443) or goat-anti-human AF647 (Invitrogen, A21445). The polyclonal rabbit anti-IE1 antibody was generated against the peptide sequence N'-KRKMDPDNPDEGPS-C'. HCMV virus

was propagated in MRC5 or NHDF fibroblast cells and the virus was isolated from infected cell supernatant and cell lysate following sonication, then further isolated by ultracentrifugation (20,000 rpm, 1.5 hours (h), 25 °C) over a 20% sorbitol cushion (Fisher Scientific, S459-500) using a SW-28 rotor (Beckman Coulter). The resulting virus was resuspended in 3% bovine serum albumin (BSA) in PBS and stored at −80 °C. The viruses were titered in ARPE-19 and either NHDF or MRC5 fibroblasts to determine the infectious units per milliliter (IU/mL).

**Mouse vaccination and hybridoma fusion**. Twelve-week-old *VelocImmune®* female mice were immunized with either 100 μg isolated HCMV (Merlin) formulated with 100μg of PolyIC (VacciGrade, InVivoGen), 50μg TB40/E and 50μg VHL/E formulated with PolyIC, or electroporated with 100μg of cDNA plasmid encoding the gH/gL TB40/E strain sequences. Protein content was determined using a BCA protein assay kit (Thermo Fisher, 23225). Following the prime immunization (100 μg), mice immunized with the purified virus were boosted 14 days post-prime and 21 days post DNA immunization. Each group received a total of four boosts and blood was collected from the submandibular vein at 50 and 150 days post-prime. Following sera neutralization analysis, three mice were selected from the DNA immunization group for hybridoma fusion and received two final boosts each consisting of 50 μg of TB40/E virus at −5 and −2 days before being euthanized by CO$_2$ asphyxiation. The spleens were processed to a single-cell suspension and hybridomas were generated. Splenocytes from these mice were fused by using polyethylene glycol with hypoxanthine-aminopterin-thymidine-sensitive F23.1 cells using the ClonalCell-HY system and protocol (StemCell Technologies, 03800)[76]. Briefly, the individual B-cell clones were grown on soft agar and selected for screening using a robotic ClonaCell Easy Pick instrument (Hamilton/Stem Cell Technology). Individual clones were expanded, and the supernatant was used to screen for binding to gH/gL. All animal studies were approved by the Icahn School of Medicine Institutional Animal Care and Use Committee (IACUC). Animal studies adhere to the ARRIVE guidelines.

**Antibody sequence analysis of heavy chain variable region and junctional diversity**. Sequencing of the variable heavy and kappa chains was obtained by using SMARTer 5' RACE technology (Takara Bio USA) adapted for immunoglobulins to amplify the variable genes from heavy and kappa chains. Briefly, RNA was extracted from each hybridoma using RNeasy Mini Kit (Qiagen, 74004), followed by first stand cDNA synthesis using constant gene-specific 3' primers (GSP1) based on the specific mouse isotype of the hybridoma and incubation with the SMARTer II A Oligonucleotide and SMARTscribe reverse transcriptrase (Takara, 634858). [GSP1 Primers (5'-3'): mG1-AGAGGTCAGACTGCAGGACA, mG2a- CTTGTCCACTTTGGTG CTGC, mG2b-GACAGTCACTGAGCTGCTCA, mG2b-GACAGTCACTGAGC TGCTCA, mcK- CCAACTGTTCAGGACGCCAT]. Amplifying PCR of the first stand cDNA product was then performed using SeqAmp DNA Polymerase (Takara, 638504) with a nested 3' primer (GSP2 Primer) to the constant genes and a 5' universal primer (kit provided) based on universal primer sites added to the 5' end during cDNA generation. [GSP2 Primers (5'-3'): mG1- CCCAGGGTCACCATGG AGTT, mG2a- GGTCACTGGCTCAGGGAAAT, mG2b- CTTGACCAGGCATC CCAGAG, mG3- GACAGGGCTCCATAGTTCCATT, mCk- CTGAGGCACCT CCAGATGTTAAC] Purified PCR product was then submitted for Sanger sequencing using 3' constant gene primers (GeneWiz, South Plainfield, NJ). Sequence results were blasted against the IMGT human databank of germline genes using V-Quest (http://imgt.org). To generate completely human anti-gH antibodies, the Fab region of the cDNA from clones 9A12, 13G1, 14E1, and 15G11 was synthesized and cloned in-framed into a pcDNA-based vector containing a human IgG1 constant region and a human kappa light chain constant region (GenScript USA Inc., Piscataway, NJ). The plasmids expressing the heavy and light chains of the respective human immunoglobulins were transfected into Expi293F cells and immunoglobulin was purified from the cell supernatant 2 days post-transfection.

**Isotyping**. Isotyping for the constant gene of the antibodies was done with the Mouse Immunoglobulin Isotyping Kit (BD, 550026) following the manufacture's protocol.

**High-throughput Expi293F gH/gL binding assay**. Expi293F cells transiently transfected with a gH/gL expressing plasmid (Lipofectamine 3000 (L3000001, Thermo Fisher)) were incubated with supernatant from the hybridoma cells 48 h post-transfection. Binding was detected using an anti-mouse IgG-AF647 antibody on a high-throughput flow cytometer (HTFC, Sartorius Group). The mean fluorescence intensity (MFI) was analyzed using FlowJo software (Tree Star, Inc.) and graphed using GraphPad Prism to create a heat map based on MFI.

**High-throughput neutralization screening of hybridoma clones**. AD169R preincubated with hybridoma supernatant (1:5) with media containing virus (total: 50 μL, MOI 0.2) was incubated at 4 °C for 1 h prior to infecting ARPE-19 and NHDF cells (10,000 cells per well) for 2 h at 37 °C. The inoculum was replaced with 100 μL complete DMEM/F-12 media and cells were fixed with 4% paraformaldehyde (PFA) and stained with Hoechst reagent (0.01 ug/mL, Molecular Probes, H3570) 18 h post-infection (hpi).

**Monoclonal antibody purification**. Monoclonals were purified by FPLC on an ÄKTA pure FPLC systems on protein G affinity columns (HiTrap-1ml, GE /Cytiva,#17-0404-01). They were dialyzed against PBS and quantitated by both BCA and OD at 280 nm.

**Virus neutralization assay**. Purified AD169R, TB40/E, TR, or Towne (MOI 0.2) virus was preincubated with diluted mAbs using either five-fold dilutions in epithelial and trophoblasts (0.016–50 μg/mL mAb) or 10-fold dilutions for fibroblasts (0.0001–100 μg/mL mAb). Preincubation occurred at 4 °C for 1 h prior to inoculum addition to ARPE-19, HTR-svNeo, MRC5, or NHDF cells (10,000 cells per well). After incubation at 37 °C, 5% CO$_2$ for 2 h the inoculum was replaced with 100 μL DMEM/F-12 media, and analyzed for infection 18–24 hpi using a plate-based imaging Celigo cytometer (Nexelcom, Version 4.1.3.0).

**HSV-1 neutralization**. Vero cells (10,000 cells/well) were infected with a HSV-1 strain (HSV-1 US-11 GFP, MOI 0.01) that expresses EGFP in place of the Us11 gene[77]. The virus preincubated with mAbs or heparin (50–0.016 μg/mL, 1 h) was added to cells and after two hours, the inoculum was replaced with DMEM. Infected cells were analyzed 24 hpi for infection using a Celigo cytometer (Nexelcom, Version 4.1.3.0). Minimum plaque surface area was defined as >5000 μm$^2$ for each condition and normalized to plaque number when HSV-1 was incubated with irrelevant influenza (IAV) antibody (PY102).

**Dissemination/plaque-reduction assay**. ARPE-19 or NHDF cells (50,000 cells/well, 24-well plate) infected with AD169R (MOI 0.01 and 0.001) incubated (1 h) with either 10 μg/mL or 0.5 μg/mL of antibody. Following a 2 h incubation, the inoculum was removed, and cells were overlaid with 1% low melt temperature sea agarose overlay for 30 min at 20 °C. After the agarose solidified, 500 μL of complete media was added to each well and placed at 37 °C, 5% CO$_2$. Media was replaced every 2–3 days throughout the experiments. On day 7 post-infection, images were collected using the brightfield and 488 nm channels of a Cytation 3 to visualize plaque numbers and approximate size for each condition using GFP as a readout for infection. On 10 dpi or 14 dpi, cells were fixed with 4% PFA and analyzed for infected cells by GFP fluorescence. TB40/E infected cells were stained with a rabbit anti-IE1 antibody.

**Flow cytometry analysis**. Cells untreated or treated with CytoFix/CytoPerm (BD, 51-2090KZ) as recommended by the manufacturer were resuspended in 1% BSA(Akron Biotech, AK8905-0100)/PBS for intact cells or 1% BSA/0.1% Saponin/PBS for fixed cells. Cells were incubated with a primary antibody (2 μg/mL) followed by a goat-anti-mouse IgG conjugated to AF647 (1:500). Cytometry data were collected using either an Intellicyte HTFC or an Attune NxT flow cytometer. The data were analyzed using Flow Jo software and graphed using Prism 8.4 software. Gating strategy and representative contour plots for each condition are provided in Supplementary Figs. 10 and 11. All conditions were repeated in technical triplicate and experimental triplicates were performed for Fusion 1 and 2.

**Competition assay**. U373-gH/gL cells (50,000) treated with CytoFix/CytoPerm were resuspended 0.1% Saponin/1% BSA/PBS and mixed with AlexaFluor-647 (AF647)-labeled antibody (1.0 μg/ml) and increasing concentrations of unlabeled antibody (5–0.01 μg/mL final concentration) at equal volumes at 4 °C for 1 h prior to analysis by flow cytometry (Figs. S10 and S11). The percent mean fluorescence intensity (MFI) was calculated using the average MFI of labeled antibody binding in the presence of a 10-fold excess of unlabeled PY102 as an irrelevant control.

**Pre/post-attachment neutralization assay**. MRC5 or ARPE-19 cells (10,000 cells/well) were preincubated with antibody (50μg/mL) followed by infection with AD169R (MOI 0.2) or virus preincubated with antibody (4 °C for 1 h). Following a 2 h infection, the inoculum was replaced with complete cell culture media, and cell infection was analyzed at 18 hpi using GFP fluorescence and anti-IE1 staining. The percent infection was calculated using the irrelevant influenza antibodies PY102 or M2E10.

**Immunofluorescence assays**. Cells were fixed (4% PFA) and permeabilized (0.3% Triton X-100 (Thermo Fisher, HFH10) in PBS before staining with a rabbit anti-IE1-1/2 antibody (0.7 ng/mL) followed by an anti-rabbit IgG-AF647 (1:1000) antibody. To quantify total cells, Hoechst reagent (0.01 μg/mL) was added to cells. Virus neutralization was quantified using a Celigo cytometer (Nexelcom Bioscience). Percent infection was calculated using ((# IE1-1+)/(# Hoechst+))*100 for each well and normalized to the percent infection using an irrelevant, non-neutralizing, or no antibody. For 15G11 staining, fixed and permeabilized AD169R-infected ARPE-19 cells were incubated with 15G11 (5 μg/mL, 1 h, 20 °C) at 6 days post-infection followed by an α-mouse Fc-AF647 detection antibody (1:500, 1 h, 20 °C). For analysis of CLIMP63 (CKAP4), cells were stained with α-CKAP4 (ENZ-ABS669-0100, Enzo Life Sciences, 0.001 μg/ml) directly conjugated to AlexaFluor 488 (AF488) using the APEX-labeling kit (Invitrogen, A10468). Cells were stained with Hoechst reagent (0.01 μg/mL) in PBS and then analyzed using the Cytation 3 Cell Imaging Multi-Mode Reader (Biotek).

**gH-AA mutant screen**. BHK cells ($5 \times 10^4$) in a 24-well plate were transfected with gH/gL pcDNA3.1 plasmid (1.5 µg) a 1:1 ratio using Lipofectamine 2000 as suggested by the manufacturer. At 48 h post-transfection, the cells were fixed with 4% PFA and stained with AF647 labeled anti-gH monoclonal antibodies (1 µg/ml) and Hoechst reagent (0.01 µg/mL).

**Antibody epitope mapping using gH microarray**. A gH peptide library compatible with the PEPperMAP Conformational Epitope Mapping service was purchased, created, and tested at PEPperPRINT (Heidelberg, Germany). A 100 µg of each monoclonal was tested on-site at PEPperPrint gH peptide library. Briefly, the sequence of gH (TB40/E strain) was elongated with neutral linkers (GSGSGSG) at the C- and N- terminus to avoid truncation of peptides before being converted into 7, 10, and 13 amino acid peptides with overlaps of 6, 9, and 12 amino acids. After peptide synthesis, a thioether linkage between the C-terminal cysteine and an appropriately modified N-terminus allowed peptides to be cyclized, resulting in a conformational gH peptide microarray library containing 1815 unique peptides printed in duplicate (3360 peptide spots). Each microarray included 128 spots for HA control peptides flanking the array (YPYDVPDYAG). Following pre-swelling and incubation with blocking buffer (Rockland MB-070, 30 min), the mAbs 1D11, 9A12, 10F8, and 15G11 were added at concentrations of 1, 10, and 100 µg/mL for 16 h at 4 °C shaking at 140 rpm. Following a wash step, each microarray copy was incubated with goat-anti-mouse DyLight 680 (0.2 µg/mL) for 45 min at room temperature, washed, and scanned using a LI-COR Odyssey Imaging System for intensities of 7/7 (red/green). The additional HA peptides lining the array were subsequently stained with mouse monoclonal anti-HA (clone 12CA5, 0.5 µg/mL) and used as internal quality control to confirm assay quality and peptide microarray integrity. Quantification of spot intensities was based on a 16-bit grayscale tiff file and microarray image analysis was performed using the PepSlide® Analyzer.

**Immunoprecipitation (IP) assay**. U373-gH/gL cells lysed with NP-40 lysis buffer [0.5% NP-40, 50 mM Tris pH 7.5, 150 mM NaCl, 5 mM MgCl$_2$, Leupeptin (2 µM), Aprotinin (2 µg/ml), phenylmethylsulfonyl fluoride (PMSF, 20 µM)] and clarified using centrifugation ($13,000 \times g$ for 5 min). The cell lysates (~$2 \times 10^6$ cells/ mL of NP-40 lysis buffer) were incubated with an antibody (5 µg) followed by Protein A-agarose (IPA3005, Repligen, Walthma, MA) for 1 h rocking at 4 °C. The agarose beads were washed ($3\times$) with NET buffer (50 mM Tris pH 7.5, 0.5% NP-40, 150 mM NaCl, 5 mM EDTA) before being resuspended with SDS-sample buffer [50 µL, 1.5% SDS, 1 M Tris pH 6.8, 50% glycerol, 600 mM DL-Dithiothreitol (DTT), bromophenol blue] and heated at 95 °C for 2 min. The proteins from the supernatant of the pelleted Protein A-agarose beads were resolved on a 10% SDS-polyacrylamide gel. The proteins were transferred to a PVDF membrane, then incubated with 10% dry milk followed by incubation with anti-gL and anti-gH antibodies (1 h at 25 °C), anti-rabbit Ig conjugated to HRP, and chemiluminescent HRP substrate (Millipore, WBKLS 0500) for visualization on a autoradiography film.

**Virus attachment**. ARPE-19 cells plated in a glass-bottom 12-well plate were infected with TB40/E UL32 eGFP virus (MOI 0.1) preincubated with α-BKV, 6E1, or 15G11 antibody (10 µg/mL, 1 h at 4 °C). Infection occurred at 37 °C for 0–2 h prior to a citrate wash buffer (40 mM Citric acid, 10 mM KCl, 135 mM NaCl, pH 3.2) for 2 min at 20 °C. Cells were washed with PBS and fixed in 4% PFA (20 min at 20 °C) and stained with Hoechst (0.01 µg/mL) stain. Plates were read using a Cytation 3 Cell Imaging Multi-Mode Reader (BioTek, Winooski, VT) and images were acquired using brightfield, UV, and fluoresent signal.

**K$_D$ determination**. Biolayer interferometry assays were performed using the Octet RED 96 instrument (SartoriusAG) to determine the association rates (k$_{on}$), dissociation rates (k$_{dis}$), and affinity (K$_D$) for each antibody. A purified monoclonal antibody was loaded onto an anti-mouse Fc IgG capture (AMC) biosensors using a constant 10 µg/mL concentration for 10 min at 20 °C. To determine the k$_{on}$, the sensors were exposed to the recombinant HCMV pentamer (strain VR1814) consisting of gH, gL, UL128, UL130, and UL131A (Native Antigen, CMV-PENT-100) starting at a concentration of 100 ug/mL (two-fold dilutions (100–6.25 ug/mL in PBS) for 3 min. The k$_{dis}$, dissociation was measured over the course of 3 min while the sensors were in PBS buffer. K$_D$ values were calculated based on the sensograms (Supplementary Fig. 12) as a ratio of k$_{dis}$/k$_{on}$. A binding model of 1:1 resulted in the best fit for each antibody and the resulting R$^2$ values are given for each antibody in Table 2.

**Statistics and reproducibility**. All statistical tests were performed using GraphPad Prism 9 software (La Jolla, CA). An asterisk identifies statistical significance and is denoted as *$p < 0.05$; **$p < 0.01$; ***$p < 0.001$; ****$p < 0.0001$. The half-maximal inhibitory concentration (IC$_{50}$) values for each monoclonal were calculated using the 4-parameter non-linear regression analysis with constraints for the bottom and the top to 0 and 100, respectively, once the antibody concentrations (x-axis) had been transformed to log scale and error bars represent the standard deviation of the mean for all relevant figures. Sample size and replicates for each experiment are listed in the figure legends. Technical replicates were prepared in parallel within one experiment and experimental triplicates were performed on separate days.

**Reporting summary**. Further information on research design is available in the Nature Research Reporting Summary linked to this article.

## Data availability

Any data not provided in the figures or Supplementary Fig.s of this study are openly available at FigShare at 10.6084 reference numbers [https://doi.org/10.6084/m9.figshare.19141949, https://doi.org/10.6084/m9.figshare.19141997, https://doi.org/10.6084/m9.figshare.19142030, 10.6084/m9, https://doi.org/10.6084/m9.figshare.19142045, https://doi.org/10.6084/m9.figshare.19142063, https://doi.org/10.6084/m9.figshare.19142102, https://doi.org/10.6084/m9.figshare.19142126, https://doi.org/10.6084/m9.figshare.19142129, https://doi.org/10.6084/m9.figshare.19142144, https://doi.org/10.6084/m9.figshare.19142153, https://doi.org/10.6084/m9.figshare.19142174, https://doi.org/10.6084/m9.figshare.19142183, https://doi.org/10.6084/m9.figshare.19152161, https://doi.org/10.6084/m9.figshare.19142165, https://doi.org/10.6084/m9.figshare.19150871, https://doi.org/10.6084/m9.figshare.19151396, https://doi.org/10.6084/m9.figshare.19151519, https://doi.org/10.6084/m9.figshare.19151924, https://doi.org/10.6084/m9.figshare.19163789][78]. Representative gating strategy and contour plots for each of the conditions shown in Fig. 3B are presented in Supplementary Figs 10–11. The original unedited western blot scans of the immunoprecipitation in Fig. 3C are also included in the Supplementary Fig. 13. Reagents from these studies are available from the corresponding author (D.T.) upon request and for research purposes only.

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

## Acknowledgements

We acknowledge Mount Sinai Innovation Partners (iP3) and Regeneron Pharmaceuticals for their support of these studies. The NIH Institutional Research Training Award—T32-AI007647, R01AI139258, R21AI147632, and RF1AG059319 partially supported A.J.P., S.O., K.S., and D.T.

## Author contributions

A.P. and D.T. designed the monoclonal antibody experiments. D.T. and T.M. designed the animal experiments and hybridoma fusion. A.P., S.O., A.D., T.K., and K.S. performed experiments and analyzed data. D.T. conceived and supervised the project. A.P. and D.T. wrote the manuscript with input from all authors.

## Competing interests

The authors declare the following competing interests: patent entitled "Anti-HCMV antibodies and antigen-binding fragments thereof" #63/240,489 has been filed by Icahn School of Medicine at Mount Sinai.
