## [Peer Review File · Communications Biology]

Reviewers' comments:

Reviewer #1 (Remarks to the Author):

The authors describe the successful generation of recombinant, human antibodies from transgenic mice against HCMV surface proteins. The amount of unique antibodies (12) was very limited, perhaps due to the limitations of the hybridoma technology. Here, a single B-cell sorting and cloning approach (which is normally used by Regeneron for the VelocImmune mice) or an immune phage display library may result in more unique antibodies. The best antibodies with nM affinity neutralize several HCMV strains in vitro and showed a good efficacy when combined with the antiviral drug Ganciclovir in comparison to antibody or Ganciclovir alone. It is unexpected, that the affinities of antibodies which have hypermutations dot have affinity in the sub nM range. The identification and the validation of the epitopes of four antibodies using a peptide array followed by binding to alanine mutants was very good.

Major Revisions:

- Recombinant, in vitro neutralizing antibodies (e.g. Nejatollahi et al 2002, Moazen et al 2016, Ohta et al 2009) or murine in vitro neutralizing antibodies (e.g. Gardner et al 2016, same group as this paper) are also described before. In my opinion, the authors do not show essential progress compared to the former publications. Showing an in vivo protection assays would be a progress which has to be shown.

Minor Revisions:

- discussion: the antibodies generated by VelocImmune mice are human antibodies, not humanized antibodies. The authors should clarify this point.

Reviewer #2 (Remarks to the Author):

The authors present successful discovery of anti-HCMV antibodies from the Velocimmune mouse platform. After an extensive screening, antibodies were studied for binding to a particular viral subunit, and also the complex, expressed on the cell surface (or in the cell for one subunit). Antibodies were found to be of high affinity. Importantly, several of them could neutralize the HCMV infection in fibroblasts, trophoblasts and epithelial cells. Their epitopes were mapped using peptide screening, and several different epitope bins were discovered, which was also confirmed with competition assays. Further, their neutralization effect on diverse clinical strains was precisely described using plaque assays pre and post attachment. In combination with ganciclovir, the candidates could limit the dissemination of virus infection. Present work is not only a report of very elegant and efficient antibody discovery, but also contributes novel knowledge on epitopes of HCMV that are relevant for neutralization and sets base for future vaccine design. To that, also clinical aspects elucidating the importance of this work are well presented in the introduction section.

A large amount of data is presented in a very systematical way, the reasoning is easy to follow and conclusions from the study very clear. Please find below the list of minor remarks which I hope you will find helpful. The language in "Methods" section is a bit too colloquial and proofing by a senior author would be helpful for the reader. Please also correct the use of standard units, and improve on Figure Legends, which sometimes do not quite match the figures.

Line 133: ÄKTA, throughout the text

Line 154: microgram, throughout the text

Line 197: microliter, throughout the text

Line 213: CO₂, O should not be a zero, please correct throughout the text

Line 227: 24-well plate (I assume) and infected on the following day

Line 228: 2 h or 2 hrs incubation

Line 229: please cite the complete incubation conditions, which medium was used etc.

Line 230: room temp? Please define the abbreviations early in the manuscript, this is laboratory colloquial language

Line 231: put back to

Line 233: Cytation 3? Is this an imaging analysis program? Please define the use and specify

further (version etc).

Line 233: please specify the protocol of fixation, or refer to further paragraphs

Line 238: 1E6 cells, please write that out

Line 240: is the abbreviation AF488 (AlexaFluor, I think) defined anywhere? Something similar comes in the line 247, please check that statement as well (only it is AF647)

Line 248: perm buffer?

Line 250: please correct the "hr" abbreviations throughout the text, these are h units

Line 251: do you mean the average response of the labelled antibody? What does 5 microgram/mL (10x) mean? I think this would be a ten-fold excess?

Line 258: please define the hpi

Line 282: Was the peptide synthesis done by authors themselves or could a source be specified? Which software was used?

Line 287: peptides

Line 302: you can leave out "1x"

Line 303: sorry, this is MgCl (2subscript)

Line 305: please define the abbreviation IP

Line 306: a million cells per preparation? Please clarify

Line 310: are you sure this was 600 mM DTT (quite a lot for common protocols)

Line 312: what does PVMF mean?

Line 316: HRP abbreviation is not defined, and this would be important as it also appears in peptide sequences as a sequence of amino acids :)

Line 319: source of plates

Line 339, IC50, 50 in subscript

Line 371: presented

Line 387: some level of reduction of the infection?

Line 440: please reword: binding demonstrated a loss of binding

Line 549: "therapy" might be understood as that in vivo data will be presented, would you consider rewording this as treatment?

Line 586: Velocimmune mice produce fully human antibodies, not humanized, which is even better

Line 589: Human antibodies

Line 589: I think you mean lower probability of immunogenicity, no need for reformatting of variable domains and hence potentially better developability – would you consider rewording?

Line 614: how would altered effector functions aid the functionality of neutralizing antibodies? These are IgG2a and devoid of most typical effector functions-please specify your thoughts.

Line 845: Figure 1 presents MRC5 as 1E

Line 895: as a cut-off

Line 902: legend for C is missing

Line 929: by all antibodies

Line 983: surely you mean kdis/kon?

Figure 5: concentrations do not match the description in the results (there it says 10 and 0.5 microg/mL)

Figure 6b: ganciclovir is ganciclovir

Reviewer #3 (Remarks to the Author):

Specific comments:

1) KD data in Table 3 need to provide sensorgrams from Octet instrument for evaluation.

2) In Figure 2, several antibody titration curves did not show a complete sigmoid curves, and IC50s listed in Table 2 will not be accurate if curve fittings are not properly conducted.

Experimental repeats are necessary for the IC50 determination.

3) Missing description and interpretation of Figure subsets: in Figure 3 (3C & 3D).

4) Supplemental Figures were not listed sequentially and follow of Figures is difficult to follow.

Useful reference published recently:

Li L, Freed DC, Liu Y, Li F, Barrett DF, Xiong W, Ye X, Adler SP, Rupp RE, Wang D, Zhang N, Fu

TM, An Z. 2021. A conditionally replication-defective cytomegalovirus vaccine elicits potent and diverse functional monoclonal antibodies in a phase I clinical trial. NPJ Vaccines 6:79.

Point-by point response:

Reviewer 1 comments:

Reviewer 1 requests that an in vivo model should be included in the manuscript.

We agree that demonstrating monoclonal inhibition is important for anti-viral therapeutics. However, evaluating the effectiveness of therapeutics against human cytomegalovirus is quite problematic because it does not naturally infect mice and using mouse CMV would not be useful for human CMV. The animal models available for human CMV are severely immunocompromised mouse models that requires implantation of human cells/tissues. Despite these models, the physiological relevance for such data is challenging given that the animal has a limited immune response and does not recapitulate human in vivo infection conditions. Also, using an immunocompromised mouse would require extensive optimization of human cell implantation and treatment options that could not be included as a single plot. In fact, the enormity of demonstrating the efficacy of monoclonal antibodies against human CMV in an immunocompromised mouse model would warrants its own manuscript. Thus, we feel that inclusion of an animal model to the study is outside the scope of the current manuscript and would be pursued in future studies. This point has been included in the Results section (lines 552-556) to demonstrate our commitment to the next phase of our studies.

Reviewer 1 comments on the essential progress of the antibodies compared to Gardner et al 2016, Nejatollahi et al 2002, Moazen et al 2016, and , Ohta et al 2009 .

We respectively disagree with this point because:

- 1) Our study has utilized novel immunization strategies in combination with high-throughput screening using multiple cell types and a virus with encoding for different gH/gL than the immunogen.
- 2) The neutralizing anti-HCMV antibodies were isolated from Velocimmune mice that were affinity matured through multiple immunizations. Further, we have generated fully human antibodies to demonstrate the effectiveness.
- 3) Our study has demonstrated neutralization of four CMV genotypes in four different cell types and demonstrated inhibition of virus proliferation.
- 4) We explored the mAb's epitope through diverse experimental approaches: competition studies, cyclic peptide array analysis, and mutagenesis studies.
- 5) We determined the affinity of the antibodies against protein complexes as compared to peptides which do not recapitulate the binding of the antibodies to virus particles.

Reviewer 1 Minor Revisions: The antibodies generated by VelocImmune mice are human antibodies, not humanized antibodies.

This is a good point. We corrected this by removing the word humanized from the text or described the antibodies as human or human/mouse chimera antibodies were appropriate. (Lines 19, 80-83, and 324).

Reviewer 1 comments mentions that it was unexpected that the affinities of antibodies with hypermutations do not have affinity in the sub nM range.

We expect that that nM scale of the KDs for our antibodies is likely due to the binding of a multi-subunit complex. Importantly, the KDs from the pool of antibodies is similar suggesting the functional of the antibodies is probably due to the targeting a specific epitope as compared to the binding constant.

Reviewer 2 comments:

Reviewer 2 states that “Methods” section is a bit too colloquial.

The Methods section has been edited to address the reviewer's comments. (Lines 108-320)

Reviewer 2 has suggested many edits throughout the manuscript.

These comments were addressed, but too numerous to specifically highlight.

Reviewer 3 comments:

Reviewer 3 comments that the sensograms that were used to determine the antibodies KD (Table 3) should be included.

We have included the sensograms as Supplemental Figure 10.

Reviewer 3 comments several antibody titration curves did not show a complete sigmoid curves, and IC50s listed in Table 2 will not be accurate if curve fittings are not properly conducted.

We have repeated the titration several times in fibroblast expanding the range of antibody concentrations and have updated Figure 2A and Table 2. The IC50 values were determined using 4-parameter non-linear regression analysis with constraints for the bottom and the top to 0 and 100 respectively once the antibody concentrations had been transformed to log scale. This was specified in the revised manuscript. (Lines 317-320).

Reviewer 3 comments that there is a missing description and interpretation of Figure subsets: in Figure 3 (3C & 3D).

We have included the description and interpretation of Figure 3C and 3D in the revised manuscript. (Lines 410-417).

Reviewer 3 comments Supplemental Figures were not listed sequentially and follow of Figures is difficult to follow.

This point has been addressed according with the Figures in ordered as described. Note, some of the supplemental figures were included to support of the data and were included at the end of Supplemental Figures.

Reviewer 3 suggested that Li et. al 2021 be included in the manuscript.

This is an excellent point and was included to support that anti-gH/gL antibodies would likely provide protection against CMV-associated diseases. (Lines 589-592)

REVIEWERS' COMMENTS:

Reviewer #1 (Remarks to the Author):

The authors answered to all of my points. I can understand their given counter arguments in the rebuttal letter.

Reviewer #2 (Remarks to the Author):

From the first round of revision:

From the first round of revision:

The authors present successful discovery of anti-HCMV antibodies from the Velocimmune mouse platform. After an extensive screening, antibodies were studied for binding to a particular viral subunit, and also the complex, expressed on the cell surface (or in the cell for one subunit). Antibodies were found to be of high affinity. Importantly, several of them could neutralize the HCMV infection in fibroblasts, trophoblasts and epithelial cells. Their epitopes were mapped using peptide screening, and several different epitope bins were discovered, which was also confirmed with competition assays. Further, their neutralization effect on diverse clinical strains was precisely described using plaque assays. In combination with ganciclovir, the candidates could limit the dissemination of virus infection. Present work is not only a report of very elegant and efficient antibody discovery, but also contributes novel knowledge on epitopes of HCMV that are relevant for neutralization and sets base for future vaccine design. To that, also clinical aspects elucidating the importance of this work are well presented in the introduction section.

A large amount of data is presented in a very systematical way, the reasoning is easy to follow and conclusions from the study very clear. Statistical analysis is OK.

Revision of the current version:

The authors have made an effort to improve the original version of the manuscript, but still I would ask them to consider the following points:

- Discussion, paragraph 1: still describes the discovery and utility of chimeric and humanized antibodies, while yours are fully human. Please correct
- Please correct μL and μg to microliter and microgram units, also in the Figures. Also 50 in EC50 should really be in subscript. I know this is a lot of work, but this is a respectable journal and your excellent data deserve so much attention.
- Figure 1: MRC5 cells are still Figure 1E, which is not included in the Figure legend
- Figure 2 legend: „across epithelial cells, trophoblast“ – please finish the sentence
- Figure 5: concentrations of the antibody still do not match the text (line 491 states : neutralizing antibodies (10 or 0.5 $\mu\text{g}/\text{mL}$))
- Figure 6: still features ganciclovir instead of ganciclovir, please correct
- Supplemental Figures 1-9 are in the text as Figure S1 etc, and the rest as Supplemental Figure 10 etc.
- Labels to Supplemental Figures 10 and 11 promise percentages of AF647-positive cells listed in the plots, but they are illegible up to a very strong magnification. Is M2E10 still a negative control in the staining shown in the Supplemental Figure 11 and if yes, why are all cells in the positive gate?
- Supplemental Figure 12 has a panel labeled A. but there is only one panel, and the legend does not mention an A. I guess rPC is recombinant pentamer complex – the abbreviation is not defined anywhere.
- Supplemental Table 1 appears twice.

Reviewer #3 (Remarks to the Author):

The revised manuscript is in a much better form than the previous one. Authors addressed most of my concern but more careful edits are needed as some simple errors still present, e.g.

- 1) Line 178, 'from' rather than 'form'
- 2) Line 454, Figure 6 or supplemental SFig6?
- 3) Authors state that 'repeated the titration several time, should included the standard error and number of repeats in the graphs (Figure 2A).

Point-by point response:

Reviewer #1:

Reviewer 1 comments” “The authors answered to all of my points. I can understand their given counter arguments in the rebuttal letter.”

No response required.

Reviewer #2

Revision of the current version:

1) Discussion, paragraph 1: still describes the discovery and utility of chimeric and humanized antibodies, while yours are fully human. Please correct

Thanks for this point. We addressed this by describing the molecular nature of the antibodies that were generated from transgenic mice as containing a human Fab region. We prefer not to describe the mAb isolated from transgenic mice as human because they continue to contain a mouse Fc region. We feel strongly about the terminology we used in the manuscript and portrays a more accurate description of antibodies in the study. As described in Supplemental Figure 9c, we demonstrate that the...

2) Please correct μ L and μ g to microliter and microgram units, also in the Figures. Also 50 in EC50 should really be in subscript.

This is an excellent point and has been completed throughout the manuscript document and all figures.

3) Figure 1: MRC5 cells are still Figure 1E, which is not included in the Figure legend

Figure 1E has been added to the Figure 1 Legend.

4) Figure 2 legend: „across epithelial cells, trophoblast“ – please finish the sentence

Completed

5) Figure 5: concentrations of the antibody still do not match the text (line 491 states: neutralizing antibodies (10 or 0.5 μ g/mL))

Completed, there was a 20-fold decrease from concentration A to concentration B

6) Figure 6: still features ganciclovir instead of ganciclovir, please correct

Corrected

7) Supplemental Figures 1-9 are in the text as Figure S1 etc, and the rest as Supplemental Figure 10 etc.

We have addressed this comment based on the checklist formatting recommendations.

8) Labels to Supplemental Figures 10 and 11 promise percentages of AF647-positive cells listed in the plots, but they are illegible up to a very strong magnification. Is M2E10 still a negative control in the staining shown in the Supplemental Figure 11 and if yes, why are all cells in the positive gate?

We have edited the figure to include a larger sized number describing the cell percentage and this can be visualized with minimal magnification. In addition, we clarified the gating strategy demonstrating the percentage of positive cells of Supplemental Figure 10 and 11 using the previously described anti-gH mAb 5C3 as a positive control (Gardner et al 2016, Nat Comm).

The anti-influenza M2 mAb M2E10 was used as an irrelevant (negative) control for the anti-gH binding assays (Figure 3b and Supplemental Figures 10 and 11). Unfortunately, we observed some M2E10 cross-reactivity only in fixed and permeabilized cells. This point is mentioned in Figure Legend 3. Despite this small increase in background binding of this irrelevant antibody, the anti-gB antibody 5A6 dramatically increased MFI for gB expressing cells only. In addition, the anti-gH antibodies 5C3 and mAbs of this study demonstrated low binding to the gB expressing cells. Thus, the M2E10 binding in these cells did not impact the results because binding of the anti-gH antibodies increased 100 fold and is consistent with non-specificity binding.

9) *Supplemental Figure 12 has a panel labeled A. but there is only one panel, and the legend does not mention an A. I guess rPC is recombinant pentamer complex – the abbreviation is not defined anywhere.*

Corrected

10) *Supplemental Table 1 appears twice.*

Corrected

Reviewer #3

1) *Line 178, 'from' rather than 'form'*

Corrected

2) *Line 454, Figure 6 or supplemental SFig6?*

Corrected

3) *Authors state that 'repeated the titration several time, should included the standard error and number of repeats in the graphs (Figure 2A).*

We modified the graph showing all replicates for Figure 2A and repeated some experiments to reduce high variability among replicates (e.g ARPE-19, AD169R Fusion 3).